# The ERK activator, BCI, inhibits ciliogenesis and causes defects in motor behavior, ciliary gating, and cytoskeletal rearrangement

Larissa L Dougherty[1,2], Soumita Dutta[2,3], Prachee Avasthi[1,2]

**MAPK pathways are well-known regulators of the cell cycle, but they have also been found to control ciliary length in a wide variety of organisms and cell types from *Caenorhabditis elegans* neurons to mammalian photoreceptors through unknown mechanisms. ERK1/2 is a MAP kinase in human cells that is predominantly phosphorylated by MEK1/2 and dephosphorylated by the phosphatase DUSP6. We have found that the ERK1/2 activator/DUSP6 inhibitor, (E)-2-benzylidene-3-(cyclohexylamino)-2,3-dihydro-1H-inden-1-one (BCI), inhibits ciliary maintenance in *Chlamydomonas* and hTERT-RPE1 cells and assembly in *Chlamydomonas*. These effects involve inhibition of total protein synthesis, microtubule organization, membrane trafficking, and KAP-GFP motor dynamics. Our data provide evidence for various avenues for BCI-induced ciliary shortening and impaired ciliogenesis that gives mechanistic insight into how MAP kinases can regulate ciliary length.**

## Introduction

### Cell division and ciliogenesis alternately use centrioles

A proper ciliary function is important for cellular signaling and development in most mammalian cells (Reiter & Leroux, 2017). Dysregulation of these antenna-like microtubule signaling/motile hubs can result in problems ranging from polydactyly to retinal dystrophy (Reiter & Leroux, 2017). Ciliary assembly and ciliary maintenance are regulated by intraflagellar transport (IFT), which controls the movement of materials into and out of the cilium (Ishikawa & Marshall, 2017).

Ciliogenesis occurs when cells exit the cell cycle (Kasahara & Inagaki, 2021). After cell division and during quiescence, the centrosome takes on a new role from serving as spindle poles for chromosome segregation to serving as basal bodies at the base of the cilium (Lattao et al, 2017). One of the two centrioles in the centrosome, the mother centriole, docks to the plasma membrane during G1/G0 to nucleate the cilium and recruit proteins involved in ciliogenesis (Azimzadeh & Marshall, 2010). In *Chlamydomonas*, two cilia begin to assemble from microtubule growth pushing from the basal bodies into the plasma membrane. This creates a protrusion, which ultimately becomes a cilium. As the cell reenters the cell cycle, the cilium will disassemble either through shedding or through axonemal disassembly before entering cell division (Patel & Tsiokas, 2021).

### The ERK pathway is tightly regulated to control the cell cycle

MAPK pathways regulate cellular stress responses, developmental phases of cells, and cell proliferation (Zhang & Liu, 2002). The Ras/Raf/MEK/ERK pathway is one of the central/core signaling pathways that has been well characterized for incorporating extracellular signals into the cell to promote cell proliferation and differentiation (Shaul & Seger, 2007). At its most basic regulation in human cells, it involves the balance of ERK1/2 phosphorylation by the kinase MEK1/2, and the phosphatase DUSP6/MKP3 (Lake et al, 2016), though there is extensive crosstalk, other proteins, and feedback loops are also involved in the regulation of this phosphorylation cascade.

### Various MAPKs have been found to regulate the ciliary structure

Prior work has shown that mutants for MAPKs can alter cilium length. For example, a type of MAPK in *Chlamydomonas*, LF4, which is a MAPK/MAK/MRK overlapping kinase, leads to longer cilia when mutated (Berman et al, 2003), as well as mutated MPK9 in *Leishmania mexicana* (Bengs et al, 2005). In photoreceptor cells, male germ cell–associated kinase (MAK) negatively regulates ciliary length to prevent degeneration (Kazatskaya et al, 2017). In *C. elegans*, MAPK15 directly regulates primary cilium formation and localization of the ciliary protein BBS7 and other proteins involved in cilium formation and maintenance. More recently, ERK7, another MAPK, has been found to regulate an actin-regulating

[1]Biochemistry and Cell Biology Department, Geisel School of Medicine at Dartmouth College, Hanover, NH, USA [2]Anatomy and Cell Biology Department, University of Kansas Medical Center, Kansas City, KS, USA [3]Department of Microbiology and Molecular Genetics, University of Texas Health Science Center at Houston, Houston, TX, USA

Correspondence: prachee.avasthi@dartmouth.edu

protein, CAP Zip, which is necessary for ciliary length maintenance in conjunction with other signaling pathways (Miyatake et al, 2015). In addition, it is known that ERK1/2 suppression with 1,4-diamino-2,3-dicyano-1,4-bis[2-aminophenylthio]butadiene, or U0126, can elongate cilia and decrease apoptosis in kidney cells (Wang et al, 2013a). Our goal in this work was to understand how the central signaling MAPK pathway can regulate ciliogenesis in addition to its role in regulating the cell cycle (Wortzel & Seger, 2011).

Here, we use *Chlamydomonas reinhardtii*, a unicellular green alga, which is an extensively used ciliary model organism with well-conserved ciliary proteins, structure, and function to humans (O'Toole et al, 2012). The ERK activator (E)-2-benzylidene-3-(cyclohexylamino)-2,3-dihydro-1H-inden-1-one, also known as BCI, induces ERK1/2 activation and ultimately alters multiple pathways that are important for ciliogenesis and ciliary maintenance. This pathway has been previously manipulated through the MEK1/2 inhibitor U0126 to prevent MAPK activation and to lengthen cilia (Avasthi et al, 2012). To understand why this process occurs, we looked at various processes that are involved in ciliary length maintenance and assembly.

# Results

## BCI-induced MAP kinase phosphorylation disrupts ciliary maintenance and assembly in *Chlamydomonas reinhardtii*

Given the lengthening effects of ERK inhibition on ciliary length through U0126 (Avasthi et al, 2012), we wanted to test the specificity of the pathway in ciliary regulation through activation of the pathway. BCI has previously been found to activate ERK1/2 upon inhibiting the dual-specificity phosphatase/MAP kinase phosphatase, DUSP6/MKP3, in addition to DUSP1 (Fig 1A). BCI was selected as a tool because of the dramatic phenotype of completely blocked ciliary growth. BCI does not inhibit various other FGF inhibitors or related phosphatases including Cdc25B, PTP1B, and Dusp3/VHR in zebrafish (Molina et al, 2009; Shojaee et al, 2015). To test whether this intended effect occurs in *Chlamydomonas*, we checked MAPK phosphorylation after 15-min intervals of treatment with BCI for up to 1 h using an antibody for human ERK1/2. Phosphorylation increased within 15 min and then decreased by 60 min, indicating that BCI activates phosphorylation of MAP kinases in *Chlamydomonas* (Fig 1B). For data throughout this study, $P < 0.05$ is considered significant.

To determine whether ERK1/2 activation through BCI treatment can shorten cilia, we tested a range of BCI concentrations and measured cilia after a 2-h treatment when length changes are typically apparent. We saw a dose-dependent effect on ciliary length where increasing BCI concentrations decreased ciliary length up to 45 $\mu$M BCI where cilia were completely resorbed (Fig 1C and D). This decreased length could be a result of reduced assembly or increased disassembly (Marshall et al, 2005). To test assembly, we severed cilia with low pH and tested ciliary reassembly (Lefebvre, 1995). In the presence of BCI, cilia could not regenerate at all (Fig 1E) confirming that there is a strong assembly defect when MAPK is activated through inhibition of its

dephosphorylating enzyme despite the presence of excess ciliary protein available for ciliogenesis (Rosenbaum et al, 1969).

We repeated these experiments in hTERT-RPE1 cells to see whether this was a *Chlamydomonas*-specific effect. Like *Chlamydomonas*, RPE1 cells showed a dose-dependent effect on ciliary shortening (Fig S1A and B). In addition, BCI induces ERK1/2 phosphorylation in RPE1 cells within 15 min of treatment (Fig S1C). Together, these data show that BCI induces phosphorylation of MAP kinases during short treatments, prevents ciliary assembly in *Chlamydomonas*, and prevents ciliary growth in RPE1 cells.

## MAP kinase phosphatases regulate ciliary length

To determine whether ciliary shortening is due to inhibitor targeting of DUSP6 in *Chlamydomonas*, we used genetic mutants of potential *Chlamydomonas* DUSP6 orthologs. To identify the *Chlamydomonas* DUSP6, we compared the zebrafish DUSP6 protein with the *Chlamydomonas* genome and took the top 3 most similar proteins to investigate which will be referred to here on as the putative DUSP6 ortholog mutants (Fig S1D and E). We first compared their BCI binding pockets. BCI is predicted to interact with human Trp262, Asn335, and Phe336, and the general acid loop backbone according to docking studies (Molina et al, 2009). The predicted zebrafish BCI binding pocket was partially conserved to the putative DUSP6 *Chlamydomonas* ortholog mutants (Fig S1D and E). We acquired these single mutants from the *Chlamydomonas* Resource Center and confirmed their mutation through PCR confirmation of a cassette insertion within the gene (Fig S1F). This cassette insertion in the target protein gene knocks out that protein. After this confirmation, these mutants were then crossed to generate double mutants. We tested both their ability to induce pMAPK expression upon treatment with BCI (Fig 1F) and their ciliary phenotypes (Fig 1G and H). Comparing steady-state ciliary lengths showed that the DUSP6 ortholog mutant *mkp2* had significantly shorter ciliary lengths when compared to WT (Fig 1G) and half as many ciliated cells (Fig S1G), though it was able to reciliate and regenerate back to previous lengths within 2 h. The *mkp3* and *mkp5* single-mutant regeneration did not significantly differ from WT, and cells were nearly fully reciliated within 2 h (Figs 1G and S1G). In addition, the single mutants increased pMAPK expression after 15 min of BCI treatment (Fig 1F). Because the dusp6 ortholog single mutants could still express pMAPK and regenerate cilia, it was possible that these phosphatases could compensate for loss of function of one phosphatase. Thus, we crossed the dusp6 ortholog single mutants together and found that not only was pMAPK signal attenuated in each double mutant and almost completely abolished in *mkp2* × *mkp3* and *mkp3* × *mkp5* mutants (Fig 1F), but also *mkp2* × *mkp3* and *mkp3* × *mkp5* mutants could not regenerate their cilia to normal or mutant lengths in 2 h (Figs 1H and S1H). Because *mkp2* × *mkp3* and *mkp3* × *mkp5* mutants recapitulated the BCI phenotype, we chose to focus on these double mutants in comparison with BCI for the rest of this study. These data support that there may be redundant functions for these MKPs in *Chlamydomonas* that can compensate for loss of function of one MAP kinase phosphatase, that MAP kinase phosphatases play a role in regulating ciliary length, and that BCI potentially targets more than one MKP in *Chlamydomonas*.

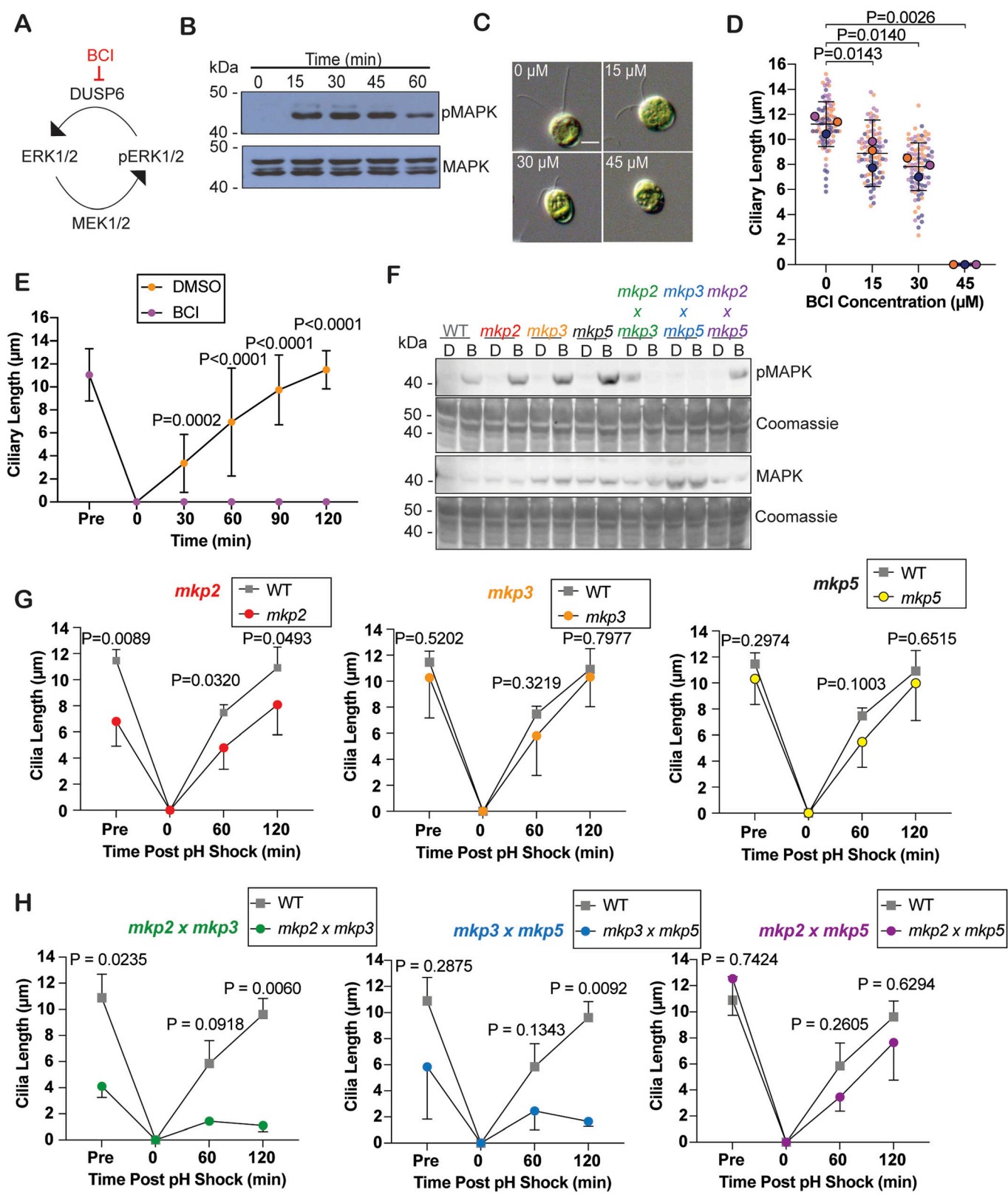

**Figure 1. BCI activates the MAPK pathway in *Chlamydomonas reinhardtii* and inhibits ciliogenesis.**
**(A)** MAP kinase ERK1/2 is phosphorylated by MEK1/2 and dephosphorylated by DUSP6. **(B)** Western blot comparing MAPK and pMAPK protein expression after a 2-h treatment with 30 μM BCI for the indicated times in *Chlamydomonas*. **(C)** Representative images of steady-state *Chlamydomonas* ciliary length in 0 μM BCI to 45 μM BCI after a 2-h treatment. Scale bar is 5 μm. **(D)** Quantification of ciliary lengths in (C). Error bars show the mean with the 95% confidence interval for the averages from each

### BCI disrupts ciliary KAP-GFP dynamics

Ciliary entry of the anterograde motor, kinesin-2, is known to be required for ciliogenesis (Adams et al, 1982). It is possible that cilia shorten and cannot regenerate in the presence of BCI because of blocked kinesin-2 entry into cilia. To test this hypothesis, we first checked whether kinesin-2 was present in the cilium and basal bodies of BCI-treated *Chlamydomonas* cells using a GFP-tagged kinesin accessory protein on kinesin-2, also known as KAP-GFP, previously generated and validated by Mueller et al (2004) (Fig 2A) (Mueller et al, 2004). The KAP subunit is a cargo adaptor protein, which is known to interact with IFT complexes and other cargoes for anterograde transport. Defects in this adaptor can have a wide range of implications for ciliogenesis. After a 2-h treatment with BCI, cells with cilia maintained KAP-GFP fluorescence at their basal bodies regardless of BCI concentration (Fig 2B). However, at 30 $\mu$M BCI, total KAP-GFP fluorescence in the cilia decreased (Fig 2C and D). Interestingly, the normalized fluorescence per micrometer of cilia was not significantly different between DMSO- and BCI-treated cells (Fig 2E), suggesting that KAP-GFP fluorescence proportionally decreases with ciliary length.

To determine how KAP-GFP dynamics respond to BCI treatment, we used total internal reflection fluorescent (TIRF) microscopy to measure KAP-GFP dynamics in real time (Fig 2F). Previous studies have shown that kinesin-2 expression and injection size are negatively correlated to ciliary length (Brown et al, 1999; Ludington et al, 2013). From kymographs of KAP-GFP movement in cilia using TIRF, we measured frequency (Fig 2G), velocity (Fig 2H), and train size (Fig 2I) between 60 and 120 min. In opposition to Ludington et al (2013), frequency continually decreased from 60 min to 120 min in BCI, whereas velocity and train size decreased by 60 min and that decrease was maintained even after 120 min in BCI-induced shortening cilia. Because we see a decrease in the frequency of KAP-GFP entering cilia, and a decreased velocity of KAP-GFP movement in cilia in addition to shortening cilia, we suggest that KAP-GFP ciliary entry is disrupted and ciliary maintenance cannot be achieved to allow proper ciliogenesis to occur under MKP-inhibited conditions.

### BCI inhibits KAP-GFP protein synthesis

Given KAP-GFP entry defects, we checked total KAP-GFP protein expression levels in regenerating cells in the presence of DMSO or BCI (Fig 3A). Interestingly, we found that total KAP-GFP protein expression in the cell was not replenished in BCI-treated cells after 2 h (Fig 3B and C), which was not significantly rescued with the proteasome inhibitor MG132 (carbobenzoxy-Leu-Leu-leucinal) (Fig S2A and B). We measured KAP-GFP intensity at the

basal bodies during these time points and found that KAP-GFP fluorescent intensity at the basal bodies was not significantly altered (Fig 3D). This indicates that KAP-GFP can still be targeted to the basal bodies despite the inability of cilia to assemble. These data indicate that BCI inhibits KAP-GFP protein expression in the cell, but this expression defect is uncoupled to the ciliary entry defect given that recruitment is unaltered.

### BCI has minor effects on a transition zone protein but not on gross transition zone structure

Given the role of NPHP4 function in the transition zone, we investigated the role of this protein in regulating ciliogenesis. The transition zone is the ciliary gate of the cell, which works to regulate protein entry and exit at the cilium (Gonçalves & Pelletier, 2017). Unchanged KAP-GFP localization at the basal bodies but decreased frequency of KAP-GFP ciliary entry suggests that there could be a defect in the transition zone structure. To test this, we first looked at this structure with electron microscopy to identify visible structural defects. Cross sections of the transition zone in both untreated and BCI-treated cells appeared unaltered, noting the classical "H-shape" and wedge connectors (Diener et al, 2015) (Fig 4A). We also checked a transition zone protein, NPHP4, in BCI-treated cells. Previous studies have found that altered NPHP4 expression decreases IFT velocity, changes protein composition in the cilium, and disrupts ciliary protein entry (Awata et al, 2014; Wang et al, 2022). In both steady-state and regenerating cells with BCI, NPHP4 still localized as two puncta at the transition zone, which supports the unaltered structure visualized with EM, though NPHP4 immunofluorescence was decreased (Fig 4B and C). Together, these data suggest that BCI does not grossly disrupt the transition zone as a mechanism for altering ciliogenesis, although it may inhibit localization of proteins at the transition zone that may ultimately affect trafficking into the cilium.

### BCI disrupts membrane trafficking

The altered KAP-GFP trafficking in the cilium and decreased NPHP4 at the transition zone with a lack of visible structural changes made us curious whether other trafficking defects were present in the cell. Membrane trafficking in *Chlamydomonas* has been found to be important for maintaining ciliary length and for ciliary assembly. Previous studies have shown that the Golgi-derived membrane is important for ciliary assembly and maintenance (Dentler, 2013), and inhibiting internalization through Arp2/3 complex perturbation also results in defective ciliary assembly and maintenance (Bigge et al, 2023). Although EM of the Golgi

---

trial (n = 30, N = 3). The *P*-value was calculated with a one-way ANOVA with Dunnett's multiple comparisons test. This super plot shows averages from each trial with larger circles plotted on top of the individual points. **(E)** Regenerating ciliary lengths after pH shock and regrowth in 30 $\mu$M BCI (purple) or 0.5% DMSO (orange) over 2 h. Error bars are the mean with the 95% confidence interval for the means from each trial (n = 30, N = 3). *P*-values for the 120-min time points were determined using a two-way ANOVA with Bonferroni's correction. **(F)** Western blots of protein from WT (CC-5325) and DUSP6 ortholog single and double mutants. Cells were treated with DMSO or 30 $\mu$M BCI for 15 min before protein collection. PVDF membranes were probed for either p42/p44 pERK1/2 or p42/p44 ERK1/2 in *Chlamydomonas*. Total protein is shown below with Coomassie blue staining. **(G, H)** Mean ciliary length measurements with the mean and 95% confidence intervals in regenerating DUSP6 ortholog single mutants and double mutants over 2 h (n = 30, N = 3). *P*-values were used to compare the averages at each time point between WT (gray squares) and the mutant in each case (circles), which were determined with a two-way ANOVA with Tukey's multiple comparisons test.

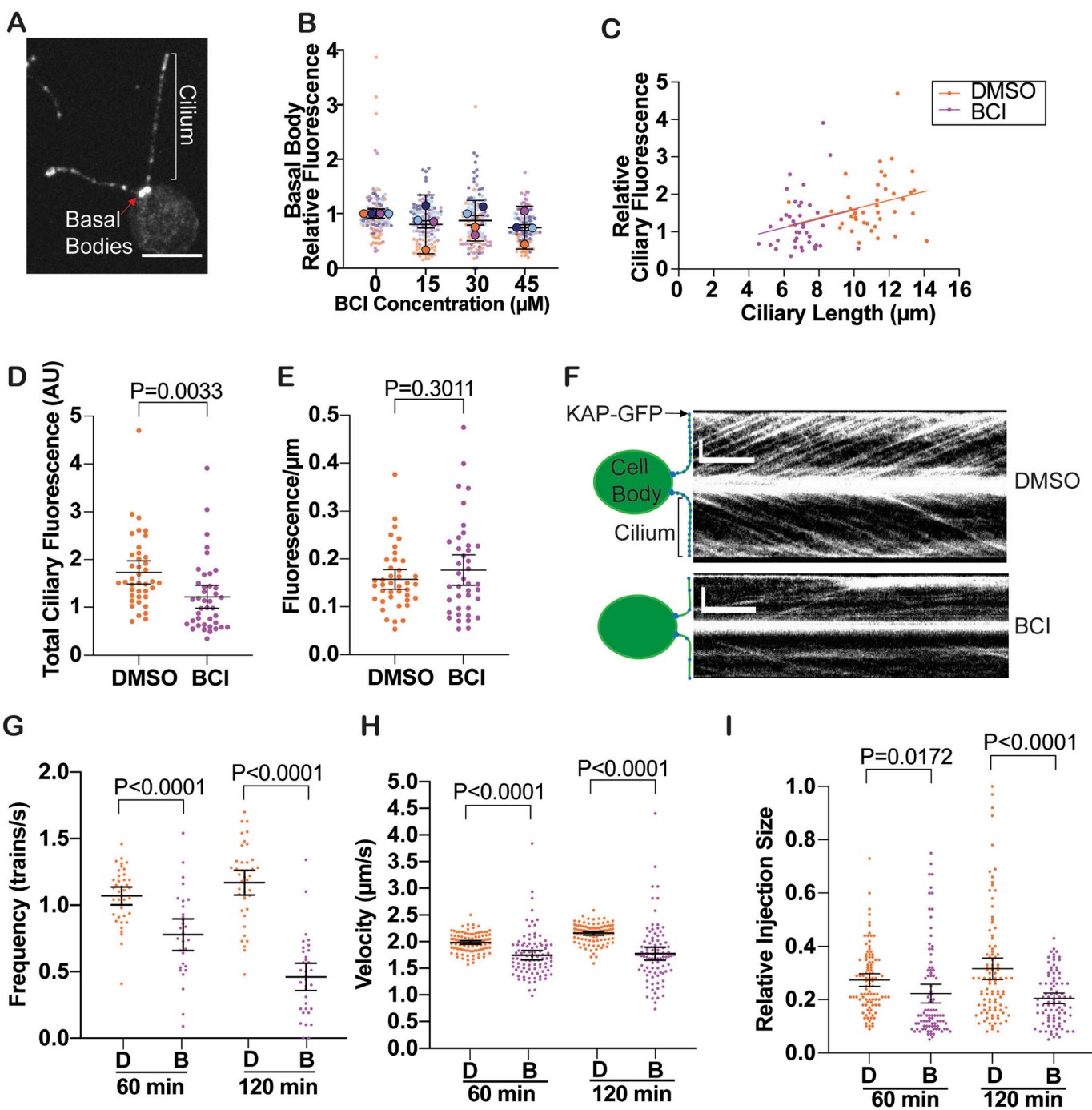

**Figure 2. KAP-GFP dynamics are altered in the cilium of BCI-treated cells.**
**(A)** *Chlamydomonas* expressing KAP-GFP. Indicated are the measured areas of KAP-GFP localization: the two basal bodies and cilia. Scale bar is 5 *μm*. **(B)** Super plot quantification of basal body fluorescence in (A) after a 2-h treatment with BCI at the indicated concentrations. Error bars are the mean with the 95% confidence interval for the averages from the trials. *P* = 0.5126, which was determined by an ordinary one-way ANOVA. **(C)** Ciliary length plotted against ciliary fluorescence for cells in DMSO (orange circles) or 30 *μ*M BCI (purple) after a 2-h treatment (n = 40, N = 1). Simple linear regression was used to generate linear regression lines and comparisons. For DMSO, y = 0.1171x + 0.4305 and r$^2$ = 0.05676. For BCI, y = 0.1174x + 0.4051 and r$^2$ = 0.2636. Comparing slopes, F = 4.048e-006 (1, 76) and *P* = 0.9984. Comparing intercepts, F = 0.005429 (1, 77) and *P* = 0.9415. **(D)** Comparison of total ciliary fluorescence shown in (C). *P*-values were determined using a *t* test. **(E)** Comparisons of ciliary fluorescence in DMSO or BCI per micrometer of cilia in (C). *P*-values were determined using a *t* test. **(F)** Example kymographs collected from total internal reflection fluorescent microscopy of KAP-GFP movement in cilia in cells treated with DMSO (top) or 30 *μ*M BCI (bottom) for 2 h. Vertical scale bars are 4 *μm*. Horizontal scale bars are 2 s. **(G, H, I)** KAP-GFP dynamics quantified from the kymographs (n = 20, N = 1). Error bars are the mean with the 95% confidence interval (n = 20, N = 1). *P*-values were calculated from a two-tailed unpaired *t* test for pairwise comparisons. **(G)** Frequency of KAP-GFP trains measured as the total amount of trains counted over the total amount of time the kymograph was collected. For DMSO, n = 40 (1 and 2 h). For BCI, n = 30 (1 h) and 34 (2 h). **(H)** Velocity of KAP-GFP trains measured as the distance traveled in *μ*m over time in seconds. For DMSO, n = 100 (1 and 2 h). For BCI, n = 93 (1 h) and 88 (2 h). **(I)** Relative injection size of KAP-GFP trains measured as the relative total fluorescent intensity of each train relative to the maximum measurement. For DMSO, n = 100 (1 and 2 h). For BCI, n = 93 (1 h) and 88 (2 h). *P*-values were determined using a *t* test.

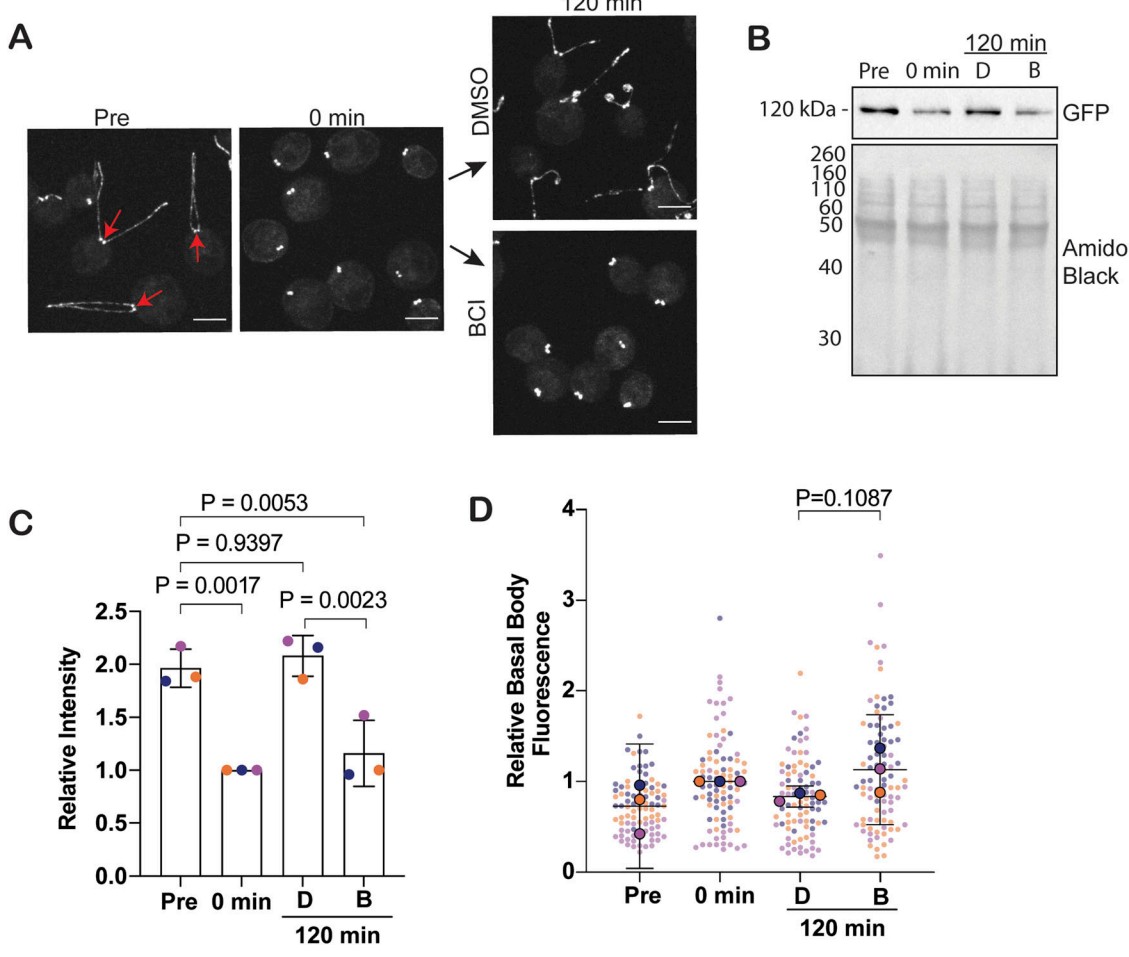

**Figure 3. BCI decreases KAP-GFP protein expression without impacting its recruitment/targeting.**
**(A)** Immunofluorescent images of KAP-GFP cells during regeneration in either DMSO or 30 µM BCI. Scale bars are 5 µm. Red arrows point to basal bodies, which are the object of quantification in (D). **(B)** Western blot for KAP-GFP expression in regenerating cells in either DMSO (D) or 30 µM BCI (B). Total protein was measured with amido black. **(C)** Quantification of (B). Error bars are the mean with SD for three independent experiments. *P*-values were calculated using an ordinary one-way ANOVA with Dunnett's multiple comparisons test. **(D)** Quantification of KAP-GFP fluorescence at the basal bodies in (A). Error bars are the mean with the 95% confidence interval for the averages from three independent trials (n = 30, N = 3). The *P*-value was calculated using an unpaired *t* test with Welch's correction.

showed no noticeable differences in Golgi structure (Fig S3A), collapsing the Golgi with BFA (Brefeldin A) intensified ciliary shortening when paired with 20 µM BCI. This suggests that BCI excludes Golgi from potential membrane trafficking–related targets of BCI (Fig 5A).

To investigate other indications of disrupted membrane trafficking, we checked the ability for cells to internalize extracellular material. Previously, Bigge et al (2023) showed that actin punctum formation, reminiscent of endocytic pits seen in yeast and are found to increase in abundance immediately post-ciliary shedding by pH shock, was absent in cells that could not regenerate cilia normally. They concluded that this is likely due to the inability of the cells to form these puncta to internalize and repurpose membrane for use in early ciliogenesis. In the presence of BCI, actin puncta increased both in steady-state cells (Fig 5B and C) and in regenerating cells (Fig S3B). However, in the steady-state *Chlamydomonas* DUSP6 ortholog double mutants, *mkp3 × mkp5* mutants, actin puncta did not significantly differ from untreated WT,

whereas *mkp2 × mkp3* mutants had significantly more actin puncta (Fig S3C). These data indicate a differential role of these phosphatases in regulating actin punctum formation. In addition, the data could also indicate that these puncta can form in BCI or *mkp2 × mkp3* mutants, but their lifetime is increased, perhaps because of incomplete endocytosis, thereby preventing membrane to be repurposed from these sites for early ciliogenesis. We measured total internalization with membrane dye uptake and found that in steady-state cells, membrane dye uptake did not differ significantly (Fig 5D and E), suggesting that BCI does not alter total internalization. Together, these data suggest that BCI disrupts ciliary membrane trafficking independent of the Golgi. Although endocytosis may be increased or slowed because of higher incidence of actin puncta detected, the overall internalization of membrane dye within the cell body is not dramatically affected. This is not wholly unexpected if the defects of plasma membrane internalization caused by BCI are preferentially impacting cilia.

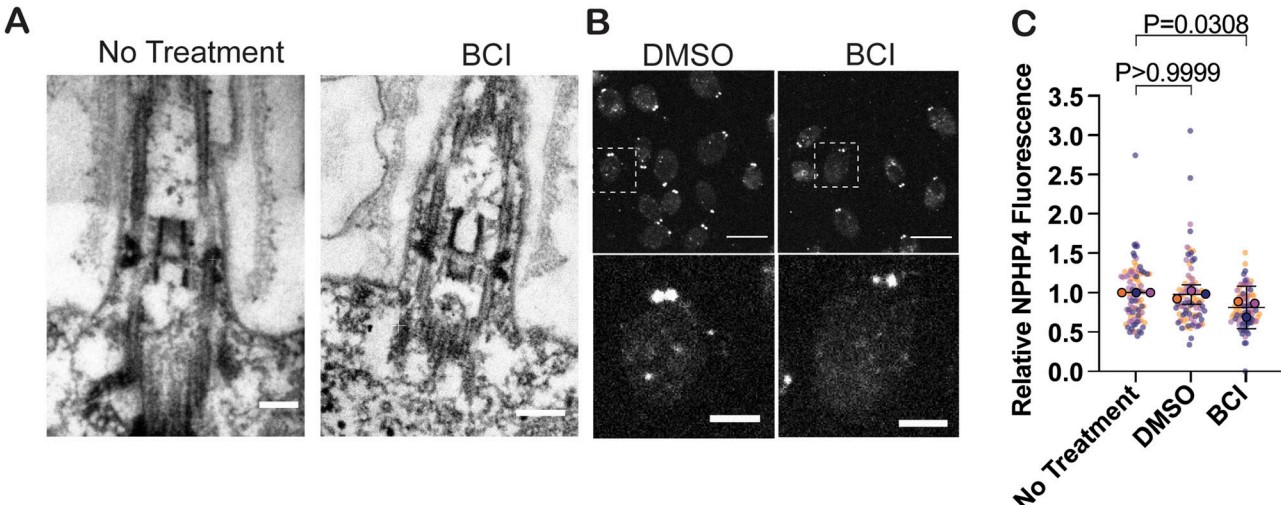

**Figure 4. NPHP4 localization is altered at the transition zone with BCI treatment.**
**(A)** Electron microscopy images of the ciliary transition zone during a 2-h steady state with or without 30 $\mu$M BCI. Scale bars are 100 nm. **(B)** Cells expressing HA-tagged NPHP4 were treated with either DMSO or 30 $\mu$M BCI for 2 h and then stained for HA. Scale bars in top images are 5 $\mu$m. Scale bars in the bottom insets are 1.5 $\mu$m. **(C)** Fluorescent quantification of the NPHP4 signal. Error bars are the mean with the 95% confidence interval for the averages from each trial (n = 30, N = 3). *P*-values were determined with an ordinary one-way ANOVA with Bonferroni's multiple comparisons test on the means from each trial.

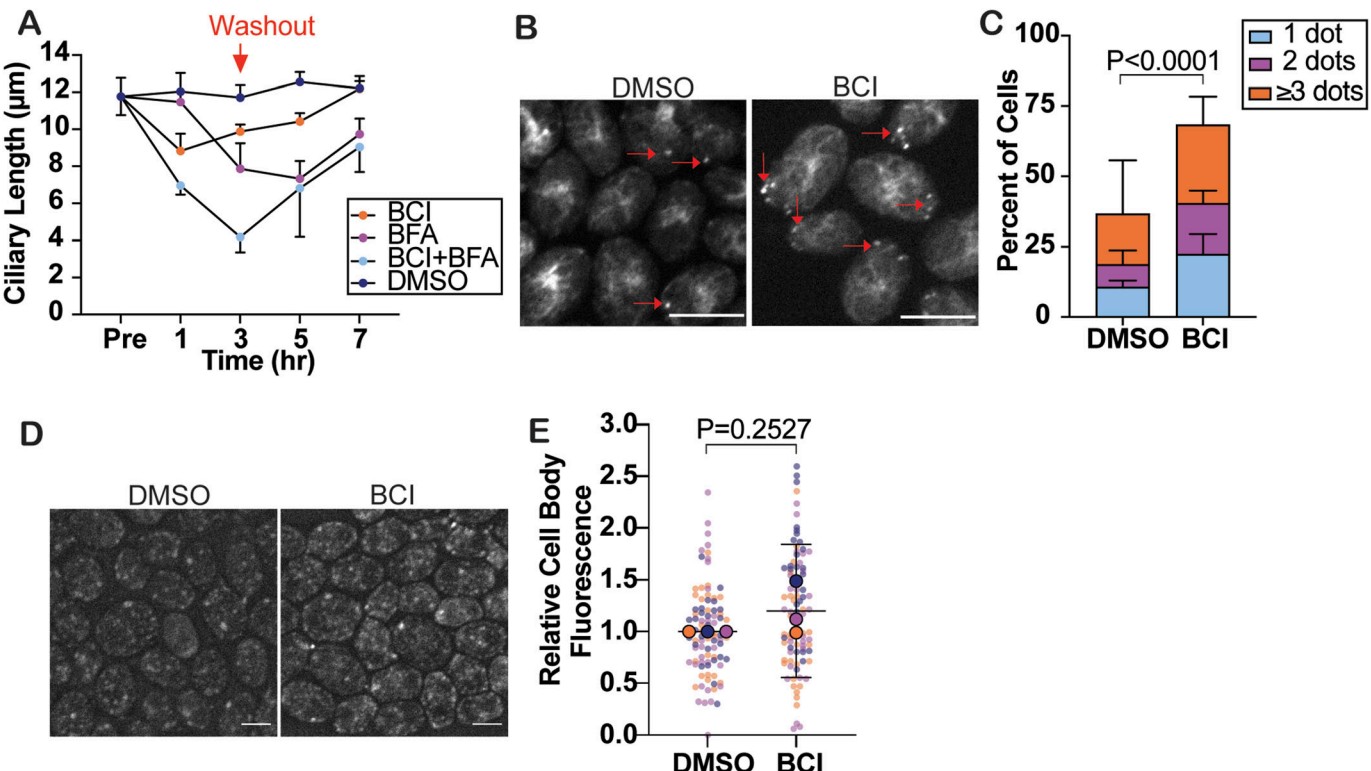

**Figure 5. Membrane trafficking is altered in BCI-treated cells.**
**(A)** Steady-state cells treated with either 20 $\mu$M BCI (orange), 36 $\mu$M BFA (purple), both BCI and BFA (light blue), or DMSO (navy) for 3 h before cells were washed out. Error bars are the mean with the 95% confidence interval for the averages from each experiment (n = 30, N = 3). *P*-values were used to compare BCI + BFA with the other treatments at 3 h and were determined using a two-way ANOVA with Tukey's correction. Comparing with BCI + BFA at the 3-h time point, *P* = 0.0117 for BCI, *P* = 0.2285 for BFA, and *P* = 0.0301 for DMSO. **(B)** Immunofluorescent images of actin puncta visualized with phalloidin in cells treated with 30 $\mu$M BCI or DMSO for 2 h. Red arrows indicate the actin puncta quantified. Scale bars are 5 $\mu$m. **(C)** Quantification of actin puncta. Error bars are the mean with SD (n = 100, N = 3). The *P*-value was determined using a chi-squared test with two-sided Fisher's exact test for cells with dots versus no dots. **(D)** Immunofluorescent images of membrane dye uptake. Scale bars are 5 $\mu$m. **(E)** Quantification of membrane dye uptake. Error bars are the mean with the 95% confidence interval for the averages from each trial (n = 30, N = 3). The *P*-value was determined with an unpaired *t* test.

**BCI disrupts cytoplasmic microtubule reorganization**

Because membrane trafficking pathways are altered with BCI, we were curious whether this defect could be due to disrupted intracellular trafficking through cytoskeleton rearrangement. We began by looking at cytoplasmic microtubule organization in regenerating cells. Upon deciliation, microtubules undergo large-scale reorganization (Wang et al, 2013b). We looked at cells regenerating their cilia and found that cells in BCI could not fully reform their microtubule structure in the presence of BCI (Fig 6A and B). To see whether cells could be forced to reorganize microtubules in the presence of BCI, we used paclitaxel (PTX), which stabilizes microtubules. In the presence of PTX and BCI, cells were able to reorganize their cytoplasmic microtubules in 60 min (Fig 6A and B), though PTX stabilization could not rescue ciliary length in BCI-treated cells (Fig 6C). To rule out the possibility that BCI induces tubulin degradation, we used MG132, which inhibits the proteasome, thereby blocking protein degradation. Cytoplasmic microtubule reorganization was not significantly rescued with MG132, suggesting that tubulin is not degraded in the presence of BCI to prevent reorganization post-regeneration (Fig 6A and B). These data suggest that BCI inhibits the mechanisms and proteins involved in cytoplasmic microtubule reorganization.

**PTX-stabilized microtubules do not rescue BCI-defective ciliogenesis**

To determine whether the defect in microtubule reorganization after deciliation blocks regeneration in BCI, we stabilized cytoplasmic microtubules with PTX to see whether the assembly defect could be rescued. After pretreating cells for 60 min with PTX to stabilize the microtubules before deciliation and then regenerating in PTX and BCI, ciliogenesis was not able to be rescued, though control cells in PTX were able to regenerate cilia normally (Fig 6D and E). These data suggest that though BCI prevents cytoplasmic reorganization, this is not the cause for immediate inhibition of ciliogenesis in regenerating cells. We checked microtubule reorganization in regenerating DUSP6 ortholog double mutants and found that *mkp2 × mkp3* and *mkp3 × mkp5* mutants had fewer cells overall, which maintained full cytoplasmic microtubule arrays, though these cells were able to reorganize microtubules to previous amounts within 60 min. This suggests a role of these ortholog mutants in regulating early cytoplasmic microtubule organization in the cell (Fig S4A and B). These data further support that BCI targets proteins involved in cytoplasmic microtubule organization, though this may not directly prevent ciliogenesis from occurring during MAPK activation.

# Discussion

In this study, we have explored mechanisms of action related to ciliogenesis for the DUSP6 inhibitor, BCI, in order to better understand how ERK can regulate ciliogenesis along with the cell cycle. Previous work has shown that silencing the MAPK pathway with MEK1/2 inhibitors can lengthen cilia (Avasthi et al, 2012; Wang et al, 2013a), and here, we show that through inhibiting the pathway

with DUSP6 inhibition, cilia cannot elongate. Although effects on disassembly remain a possibility, the striking inability to increase from zero length upon deciliation and the effects on anterograde IFT through the TIRF microscopy assays suggest a potent effect on assembly (Fig 2).

By dissecting various processes related to stages of ciliogenesis including ciliary protein synthesis, intracellular membrane and protein trafficking to basal bodies, ciliary gating, and IFT dynamics, we find it likely that activation of the ERK pathway via MKP inhibition has multiple touchpoints that ultimately affect ciliogenesis. The broad range of these effects is somewhat expected considering the quantity and diversity of known ERK targets within cells. We have several possible explanations based on the data to explain why cilia cannot grow in the presence of BCI (Fig 7). Although KAP-GFP can still be recruited to the basal bodies, its entry is inhibited in regenerating cilia in the presence of BCI, and we have shown that its synthesis is significantly decreased. These data suggest that (1) there could be modifications to existing KAP-GFP in BCI treatment that prevent its entry in ciliogenesis and that slow its kinetics in the cilia and/or (2) the transition zone is altered to prevent KAP-GFP entry.

NPHP4, a protein with an important function for ciliary gating, has decreased localization at the transition zone in the presence of BCI. Although localization is altered at a concentration of BCI, which shortens cilia partially, the physical structure is unaltered (Fig 4). The structure could be altered on a finer scale that is not detectable with the techniques employed in this work. The decrease in NPHP4 suggests that there are likely other proteins decreased at the transition zone in a BCI-dependent manner such as proteins including NPHP1 with which it directly interacts (Gonçalves & Pelletier, 2017). These proteins may be involved in directly regulating which proteins can enter and exit the cilium, and their absence may slow any movement of proteins into and out of the cilium. It is also possible that MAPK activation results in a phosphorylation cascade, which regulates NPHP4 modification, such as through direct phosphorylation, to regulate its localization rather than its protein synthesis. Previous work has shown that phosphorylation of the transition zone protein Tctex1 can promote ciliary disassembly and cell cycle progression (Li et al, 2011). Expression or localization of other proteins in the transition zone may be altered under these conditions.

In addition, we found that ERK activation through BCI causes a membrane trafficking defect. Previous work has shown that inhibition of the Golgi-derived membrane induces ciliary shortening through Golgi collapse (Dentler, 2013). The epistatic ciliary shortening in BFA and BCI together suggests that BCI may create an additive effect on the Golgi that may disrupt protein and membrane trafficking for ciliogenesis from a source separate from the Golgi, which is supported by the defects in endocytosis as measured by actin puncta. Although the electron microscopy images of the Golgi in BCI did not appear altered, there could be defects that are not visible with this method.

Our data also provide evidence that although BCI inhibits reestablishment of cytoplasmic microtubules that are typically disrupted upon ciliary regeneration, PTX stabilization of these microtubules does not rescue this defect, suggesting that the BCI ciliogenesis defect is not caused by impaired intracellular

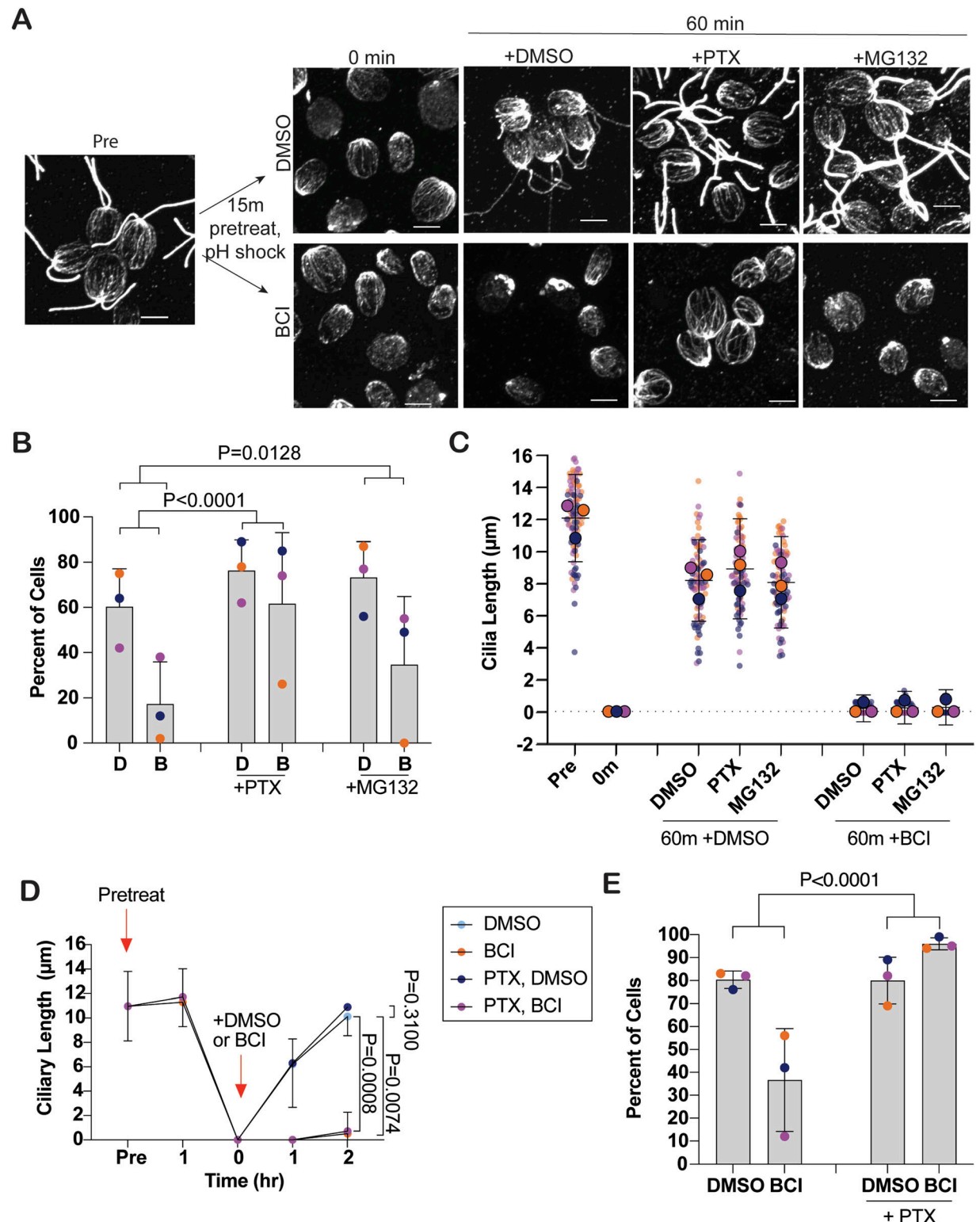

**Figure 6. PTX-induced cytoplasmic microtubule stabilization does not rescue ciliogenesis in the presence of BCI.**
**(A)** Cells were pretreated with either DMSO or 20 $\mu$M BCI for 15 min and then regenerated in DMSO or 20 $\mu$M BCI with the addition of DMSO, 15 $\mu$M paclitaxel (PTX), or 50 $\mu$M MG132 for 60 min before fixing and staining for $\beta$-tubulin. Scale bars are 5 $\mu$m. **(B)** Quantification of fully reorganized microtubules in (A) at 60 min. Bars represent the mean of the percent of cells with organized microtubules. Error bars are SD (n = 100, N = 3). D is DMSO, and B is BCI. *P*-values were determined using two-sided Fisher's exact test. **(C)** Quantification of ciliary lengths of cells in (A) pretreated with DMSO or 20 $\mu$M BCI. The averages from each trial are plotted on top of the individual data points. Error bars are the mean with the 95% confidence interval for the mean from each trial (n = 30, N = 3). **(D)** Cells were pretreated with DMSO or 15 $\mu$M PTX for 60 min and then

trafficking on microtubules (Fig 6). Impaired microtubule reestablishment may reduce paths for KAP-GFP dissipation or cycling on microtubules, trapping it at basal bodies (Fig 3A). Altered cytoplasmic microtubule dynamics may similarly affect additional proteins needed for ciliary assembly, resulting in the ultimate ciliogenesis defect in BCI. Lack of these protein highways could also prevent mRNA from localizing to its intended area for adequate protein synthesis of precursor proteins required for ciliogenesis. Constitutive ERK activation could also induce negative feedback loops that prevent transcription as well, which could inhibit mRNA formation.

Previous work has provided evidence that microtubule dynamics are required for ciliary assembly in *Chlamydomonas* and destabilization was proposed to be for the purpose of freeing up tubulin for ciliary assembly (Wang et al, 2013b). However, microtubule stabilization with PTX before deciliation and the subsequent normal ciliary regrowth suggest that microtubule destabilization is not needed for ciliogenesis (Fig 6D and E). In this study, we used a concentration of PTX (15 $\mu$M) that was just low enough to induce stabilized microtubules. It has been shown that lower concentrations of PTX induce G1/G2 arrest, whereas higher concentrations induce mitotic arrest (Giannakakou et al, 2001). Similarly, we could be seeing a differential effect for PTX on ciliogenesis where lower concentrations allow ciliogenesis to continue despite the processes induced in parallel with microtubule stabilization. In addition, although the percentage of cells with intact microtubule arrays were reduced in double mutants ("pre" in WT cells versus "pre" in double mutants) (Fig S4B), microtubules in double mutants were able to reorganize to their pre-deciliation levels in double mutants. Although microtubule integrity may be globally affected by impaired MKP function, cytoplasmic microtubule reorganization after deciliation is not affected to the degree seen for BCI treatment. This suggests the BCI has additional effects beyond what is recapitulated by these specific MKPs.

Although BCI does not inhibit various other FGF inhibitors or related phosphatases including Cdc25B, PTP1B, and Dusp3/VHR in zebrafish, in *Chlamydomonas*, our various experiments identify effects of BCI not entirely recapitulated by single or even double MKP mutants. This suggests there are likely additional MKP or off-target effects of the drug.

Together, we have found mechanisms contributing to ciliary defects, which could be the foundation of various ciliary diseases. The cilium acts as a major signaling hub in the cell to regulate cellular activities. When this structure cannot form and function properly, important signals cannot be transmitted throughout the cell. Cilia are normally resorbed for division and then reassembled during G1/G0. BCI could arrest cells at the interphase checkpoint or induce cell cycle arrest, which could inhibit protein synthesis to shift ciliary assembly to disassembly. As shown in Fig 1F, the double ortholog mutants mkp2 × mkp3 and mkp3 × mkp5 do not respond to BCI with an increase in pMAPK expression unlike mkp2 × mkp5 and

the ortholog single mutants. It has previously been shown that although pMAPK is not necessary for cell division in *Drosophila*, it is required for cell growth (Majumdar et al, 2010). Without this signal being generated, these cells may be stuck in cell division, which does not allow for ciliogenesis. Constant ERK activation can also lead to negative feedback to regulate this signal for cell cycle progression (Fritsche-Guenther et al, 2011). The ERK pathway has not been fully elucidated in *Chlamydomonas*, but it is possible that these similar mechanisms are in place for MAPKs. Further work is needed to identify the precise cause of completely inhibited cilium growth from zero length. Regardless, the insight provided here can help to provide a basis for understanding more broad mechanisms of how ciliopathies or manipulated cilia can drastically alter the fate of the cell.

# Materials and Methods

### Strains and maintenance

The WT strain (CC-5325), NPHP4-HAC, KAP-GFP, *mkp2* mutant (LMJ.RY0402.168348), *mkp3* mutant (LMJ.RY0402.191934), and *mkp5* mutant (LMJ.RY0402.083264), and Arl6 strain were acquired from the *Chlamydomonas* Resource Center (Li et al, 2019). *Chlamydomonas* cells were grown on 1.5% Tris–acetate–phosphate (TAP) agar plates under constant light (helloify BR30 LED Plant Grow Light Bulb, 9 W, 14 $\mu$mol/s blue and red light). Cells were inoculated in liquid TAP and grown under constant light and agitation for 18 h before experiments. A list of strains, antibodies, and oligonucleotides used can be found in Table 1.

The human TERT-RPE1 cells were a gift from Chris Shoemaker at Dartmouth College, Hanover, NH. These cells were maintained in DMEM (MT10013CV; Corning) with 10% FBS and 10 $\mu$g/ml Hygromycin B (InvivoGen) at 37°C in 5% $CO_2$. To induce ciliogenesis, cells were grown to 80% confluency and then serum-starved in DMEM with 0.5% FBS for 48 h at 37°C in 5% $CO_2$ before experiments.

### Crosses

Crossings were carried out according to Tulin (2019) with slight modifications. *Mkp2*, *mkp3*, and *mkp5* mutants were each mated with a mating type–positive KAP-GFP strain to generate positive mutants. Screened positive mutants were then crossed to the original single mutants. Briefly, strains were separately placed in M-N for 8 h under constant illumination, combined, and grown at a slant overnight under constant light. The pellicle was pipetted onto 4% agar plates, spread, and kept dark for at least 5 d. Zygospores were moved onto 1% tap plates and let split into tetrads overnight before being split, grown, and screened for genotypes and ciliary phenotypes.

regenerated with or without PTX with the addition of either DMSO or 20 $\mu$M BCI for 2 h. Graphed is the ciliary length quantification. Error bars are the mean with the 95% confidence interval for the averages from each trial (n = 30, N = 3). *P*-values were determined for the 2-h time points using a two-way ANOVA with Tukey's correction. **(E)** Quantification of recovered $\beta$-tubulin reorganization at 2 h in DMSO or 20 $\mu$M BCI with or without 15 $\mu$M PTX. Error bars are the mean with SD (n = 100, N = 3). The *P*-value was determined using two-sided Fisher's exact test.

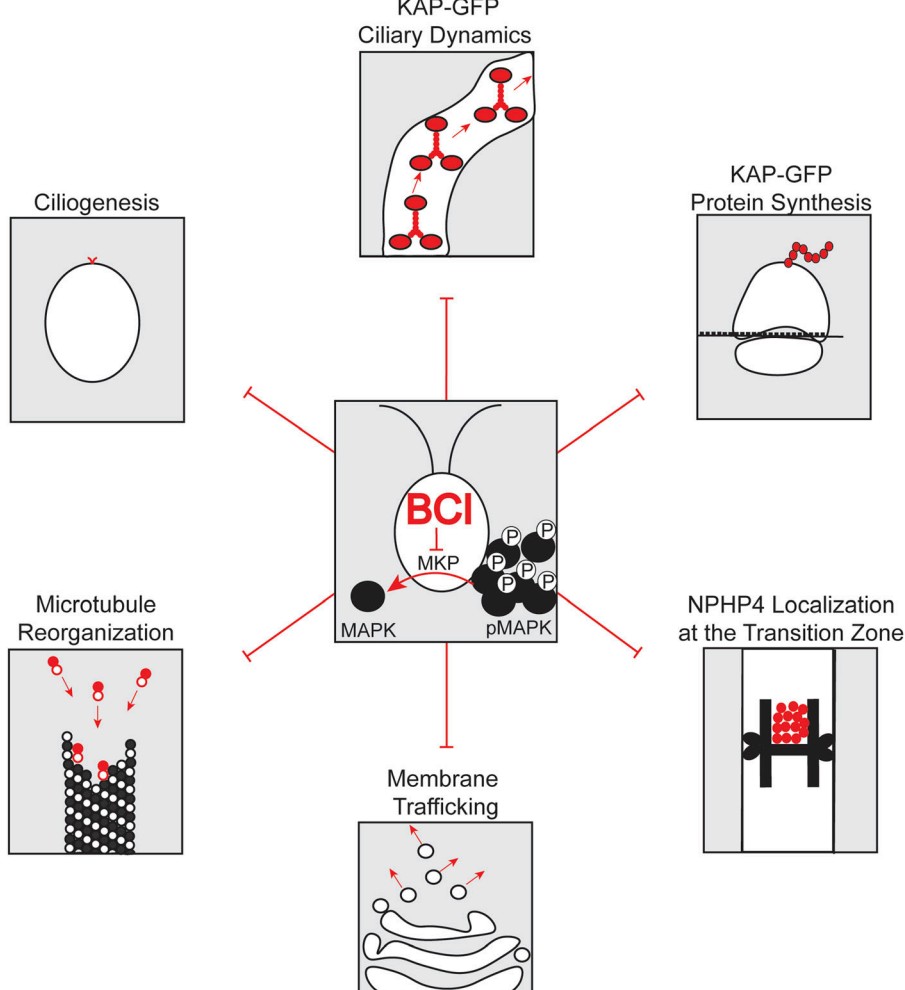

**Figure 7. Summary of BCI effects in the cell.**
BCI inhibits phosphate removal from MAPK. This ultimately alters or inhibits ciliogenesis, KAP-GFP dynamics in cilia, KAP-GFP protein synthesis, NPHP4 protein localization at the transition zone, membrane trafficking, and microtubule organization.

## Genotyping

A single colony of *Chlamydomonas* cells were placed in Chelex beads, boiled at 90°C, and then spun down at 550*g*, 1 min. This template was used along with 1× DreamTaq Buffer (Invitrogen), 2 mM dNTPs (NEB), 0.5 μM forward and reverse primers (IDT), and DreamTaq Polymerase (Invitrogen). PCR parameters were used according to the *Chlamydomonas* Resource Center. DNA was mixed with 6× loading dye to 1× (Invitrogen), run on a 1% agarose gel in TBE with 1× SYBR Safe DNA Gel Stain (S33102; Invitrogen), and imaged on a Bio-Rad ChemiDoc MP.

## Primers

*mkp2* mutant forward primer: CTGCGGACATCAGCTCAAT.
*mkp3* mutant forward primer: CAAGAGCACCTGGCACAGGAG.
*mkp5* mutant forward primer: TCGTGACAGACCTGCAGAG.
CIB1 reverse primer: CCGAGGAGAAACTGGCCTTCT.

## Ciliary length experiments

Steady-state cells were grown in liquid TAP under constant light and agitation for 18 h and then treated with >1% DMSO or BCI (B4313-5MG; Sigma-Aldrich) for 2 h. Samples were collected in equal volumes of 2% glutaraldehyde and allowed to settle before imaging. To measure cilia, 3 μl of settled cells was placed directly on a slide with a coverslip and imaged at 40× magnification with a Zeiss Axioscope 5 DIC microscope and Zeiss Zen 3.1 software (blue edition). One cilium per cell was measured with the segmented line tool and fit spline for 30 cells per time point.

Regenerating cells were grown in liquid TAP under constant light and agitation for 18 h, then pH-shocked for 45 s with 0.5 M acetic acid to bring the culture pH = 4.5 and then brought back up to pH = 7.0 with 0.5 M potassium hydroxide. These cells were immediately centrifuged at 550*g* for 1 min and supplied new TAP with or without drugs. Samples for DIC imaging were collected in equal volumes of 2% glutaraldehyde.

**Table 1.  Tools and reagents.**

| Reagent | Source | Identifier |
|---|---|---|
| Strains | | |
| Wild type | *Chlamydomonas* Resource Center | CC-5325 |
| *mkp2* | *Chlamydomonas* Resource Center | LMJ.RY0402.168348 |
| *mkp3* | *Chlamydomonas* Resource Center | LMJ.RY0402.191934 |
| *mkp5* | *Chlamydomonas* Resource Center | LMJ.RY0402.083264 |
| NPHP4-HAC | *Chlamydomonas* Resource Center | CC-5115 |
| KAP-GFP | *Chlamydomonas* Resource Center | CC-4296 |
| RPE1 cells | Gift from Chris Shoemaker, ATCC | CRL-4000 |
| Antibodies | | |
| DAPI | Biotium | 40043 |
| Rat anti-HA (for NPHP4) | Sigma-Aldrich | 11867423001 |
| Atto 488–conjugated phalloidin | Sigma-Aldrich | 49409-10Nmol |
| Phospho-p44/42 MAPK (Erk1/2) (Thr202/Tyr204) | Cell Signaling Technology | 9101 |
| p44/42 MAPK (Erk1/2) | Cell Signaling Technology | 9102 |
| Membrane stain (FMTM 4-64FX, fixable) | Thermo Fisher Scientific | F34653 |
| β-Tubulin antibody | Cell Signaling Technology | 2146S |
| GFP (D5.1) rabbit mAb | Cell Signaling Technology | 2956S |
| SYBR-Safe DNA Gel Stain | Invitrogen | S33102 |
| Alexa Fluor 488 goat anti-rat IgG | Invitrogen | A11006 |
| Alexa Fluor 488 goat anti-rabbit IgG | Invitrogen | A11008 |
| Alexa Fluor 488 goat anti-mouse IgG | Invitrogen | A11001 |
| Goat anti-rabbit IgG HRP conjugate | Invitrogen | G21234 |
| Oligonucleotides | | |
| *mkp2* mutant forward primer: CTGCGGACATCAGCTCAAT | This study | N/A |
| *mkp3* mutant forward primer: CAAGAGCACCTGGCACAGGAG | This study | N/A |
| *mkp5* mutant forward primer: TCGTGACAGACCTGCAGAG | This study | N/A |
| CIB1 reverse primer: CCGAGGAGAAACTGGCCTTCT | This study | N/A |

## TIRF microscopy and quantification

Cells were treated with 0.5% DMSO or 30 $\mu$M BCI for 2 h, and staggered to allow time for imaging at both 1 h and 2 h of treatment. KAP-GFP cells were placed on a poly-lysine–treated coverslip and diluted 1:100 in TAP. Cells were imaged with a 100× oil immersion objective on a Nikon Eclipse Ti microscope with two Andor 897 EMCCD cameras using the 499-nm laser line. Data were collected and kymographs generated in NIS-Elements software.

Quantifications were carried out on the kymographs in FIJI. KAP-GFP trains (groups of KAP-GFP molecules) are indicated by the fluorescing slants in the kymograph. For frequency measurements: Frequency = (number of trains) × (Frames per second/total frames measured). For velocity: Velocity = (tan$\theta$ in radians of train relative to the basal body) × (Frames per second) × (um/pixel conversion). For injection size (fluorescent intensity of a train), kymographs were background-subtracted with a rolling ball radius and a line thick enough to cover the largest GFP signal was drawn over each train from the basal bodies to the tip of the cilium and then normalized.

## Immunofluorescence and quantification

### KAP-GFP immunofluorescence and quantification
Cells were adhered to poly-lysine–coated coverslips for 2 min and then permeabilized in cold (–20°C) methanol for 10 min, replacing the methanol once after 5 min. Coverslips were allowed to dry, then mounted with Fluoromount-G (Thermo Fisher Scientific), and imaged with a 100× oil immersion objective on a spinning disk confocal microscope (Nikon Eclipse Ti-E with Yokogawa 2-camera CSU-W1 spinning disk system and Nikon LU-N4 laser launch) with the 488-nm laser, and images were collected with NIS-Elements software. To quantify basal body fluorescence and ciliary fluorescence, summed slice projections were collected using FIJI for each image stack and background-subtracted with a rolling ball radius. An equal-size circle was drawn around the basal bodies

across one trial, and the area, mean gray value, and integrated density measurements were collected for each time point or treatment group. The total cell fluorescence was calculated using the calculation for corrected total cell fluorescence (CTCF = IntDen–([Area of selection] × [background mean gray value])). Background mean gray values were calculated by taking the average of three random background measurements per image. For ciliary measurements, the selection area involved drawing a tight area around the cilia for one cilium per cell per time point or treatment. The length of the cilia was measured alongside the fluorescence measurements and plotted against each other (Fig 2C).

### Microtubule immunofluorescence and quantification

Staining was performed similar to Wang et al (2013b). Cells were adhered to poly-lysine coverslips, fixed in microtubule buffer (30 mM Hepes [pH = 7.2], 3 mM EGTA, 1 mM MgSO$_4$, and 25 mM KCl) containing fresh (<1 mo opened) 4% PFA for 5 min, incubated in microtubule buffer with 0.5% NP-40 for 5 min, and then permeabilized in cold (−20°C) methanol for 5 min. Coverslips were placed in a humidified chamber and blocked in 5% BSA and 1% fish gelatin for 30 min and 10% normal goat serum in block for 30 min. Coverslips were then incubated overnight at 4°C with primary antibody (β-tubulin, 2146S, 1:100; CST) in 20% block in 1× PBS. Washes were done with 1× PBS for 3 × 10 min, and coverslips were placed back in a humidified chamber with a secondary antibody (Alexa Fluor 488 goat anti-rabbit IgG, A11008, 1:500; Invitrogen) for 1 h, covered. Coverslips were washed for 3 × 10 min, covered, and allowed to dry completely before mounting with Fluoromount-G and imaging on a spinning disk confocal microscope. For microtubule quantification, 100 cells in total were counted and assigned to have either cytoplasmic microtubules or none. Cells were counted using the cell counter plugin in FIJI.

### Phalloidin staining

Phalloidin staining was performed according to Craig and Avasthi (2019). Cells were adhered to coverslips for 2 min and were fixed with fresh 4% PFA in 7.5 mM Hepes in 1× PBS for 15 min, washed in 1× PBS for 3 min, permeabilized in cold (−20°C) 80% acetone for 5 min and then 100% cold acetone, and allowed to air-dry. Coverslips were rehydrated in 1× PBS for 5 min and incubated with Atto 488–conjugated phalloidin (49409-10Nmol; Sigma-Aldrich) for 16 min, and then washed in 1× PBS for 5 min, covered (Craig & Avasthi, 2019). Coverslips were air-dried, mounted with Fluoromount-G, and imaged at 100× on the spinning disk confocal microscope. For dot quantification, maximum-intensity projections were generated for each image stack and dots per cell counted using FIJI's cell counter plugin.

### Membrane staining

Membrane staining was performed according to Bigge et al (2023). Cells were treated with 0.5% DMSO or 30 µM BCI for 2 h and then adhered to coverslips for 2 min. Coverslips were moved to ice and 200 µg/ml membrane dye (FMTM 4-64FX, fixable; F34653; Thermo Fisher Scientific) for 1 min, covered, fixed with 4% PFA in 1× HBSS for 10 min, and washed for 3 × 3 min in 1× PBS (Bigge et al, 2023). Coverslips were air-dried, mounted to slides with Fluoromount-G,

and imaged on the spinning disk confocal microscope. For quantification, summed slice projections were created and processed as described in KAP-GFP staining. The area selection measured encompassed an entire cell for 10 cells per image, three images per treatment and time point, across three trials.

### RPE1 cilium staining

Staining was performed according to Alsolami et al (2019). Cells were seeded on chamber slides and grown to 80% confluency before serum starving in DMEM with 0.5% FBS for 48 h. Media were removed, and cells were washed with 1× PBS, fixed in cold (−20°C) methanol for 10 min, and washed once more in 1× PBS for 5 min. Cells were blocked in 1× PBS with 0.2% Triton X-100 (PBX) at 37°C for 5 min and then in PBX with 3% BSA for 30 min at 37°C. Slides were moved to a humidified chamber with primary antibody (anti-acetyl-alpha-tubulin, clone 6-11B-1, MABT868; Sigma-Aldrich) diluted in PBX (1:250) added for 1 h at room temp. Slides were washed for 3 × 5 minutes in PBS and placed back in a covered humidified chamber. The secondary antibody (Alexa Fluor 488 goat anti-mouse IgG, A11001; Invitrogen) was diluted in PBX (1:500) and added to coverslips for 1 h at room temp. Slides were washed for 3 × 5 min in PBS, mounted with Fluoromount-G, and imaged with 60× and 100× oil immersion objectives on the spinning disk confocal microscope (Alsolami et al, 2019). Maximum-intensity projections were created in FIJI for each image, and primary cilia were measured using the segmented line tool for the acetylated tubulin staining.

### SDS–PAGE and immunoblotting

RPE1 cells were washed with cold 1× HBSS and then placed on ice with RIPA buffer + 10× phosphatase inhibitors for 20 min with agitation after 10 min. Cells were scraped and placed into a tube and spun at 21,000g, 10 min, 4°C. The lysate was mixed with 4× loading buffer (Invitrogen) and BME to 10%, boiled at 95°C for 10 min, and run on a 10% Bis–Tris gel for 2 h before transferring to a PVDF membrane and blocked for 1 h in TBS-T and 5% BSA. The primary antibodies pMAPK (Phospho-p44/42 MAPK [Erk1/2] [Thr202/Tyr204] Antibody #9101; CST) and MAPK(p44/42 MAPK [Erk1/2] Antibody #9102; CST) used for both RPE1 and *Chlamydomonas* were diluted 1:1,000 in TBS-T and incubated separately with blots overnight at 4°C. Blots were washed three times in TBS-T and incubated with a secondary antibody (goat anti-rabbit IgG HRP conjugate, G21234; Invitrogen) diluted 1:5,000 for 1 h at room temp. Blots were washed for 3 × 10 min in TBS-T, then incubated with Pico Chemiluminescent Substrate (Invitrogen), and imaged on a Bio-Rad ChemiDoc MP.

*Chlamydomonas* cells were centrifuged at 550g for 1 min. The supernatant was removed and replaced with 100 µl of glass beads (425–600 µm) and 100 µl lysis buffer (1% NP-40, 9% TAP, 5% glycerol, 1 mM DTT, and 1× protease and phosphatase inhibitors per 1 ml of cells spun down). Cells were bead-beating for 3 × 1 min with 1-min breaks on ice and then centrifuged at 21,000g for 15 min at 4°C. The supernatant was collected and mixed with DTT and LDS Sample Buffer (Invitrogen), and boiled at 70°C for 10 min. Protein was run on 10% Bis–Tris gels (Invitrogen) at 150 V and transferred onto a nitrocellulose or PVDF membrane.

To probe for KAP-GFP, blots were incubated with 5% milk and then probed with GFP antibody (1956S; CST) diluted at 1:1,000 in 1% milk + 1% BSA overnight at 4°C. Blots were washed for 3 × 10 min in PBS-T and incubated with a secondary antibody (goat anti-rabbit IgG HRP conjugate, G21234; Invitrogen) diluted at 1:5,000 in 1% milk + 1% BSA for 1 h at room temp. Blots were washed for 3 × 10 min in TBS-T, then incubated with Pico Chemiluminescent Substrate (Thermo Fisher Scientific), and imaged on a Bio-Rad ChemiDoc MP.

### Electron microscopy

Electron microscopy was performed by Radu Stan according to a protocol by Dentler & Adams (1992). WT cells were treated with 30 $\mu$M BCI or 0.5% DMSO for 2 h and then diluted in equal volumes of fresh, EM-grade 2% glutaraldehyde diluted from 16% in water (Electron Microscopy Sciences). Cells were settled at room temperature, and the supernatant was removed; cells were resuspended with 1% glutaraldehyde with 50 mM sodium cacodylate overnight at 4°C. The supernatant was removed, and cells were spun for 3 × 1 min at 600$g$ with 50 mM sodium cacodylate washes, then fixed, and imaged with a Helios scanning electron microscope 5CX with a STEM3+ detector (Dentler & Adams, 1992).

### Statistical analysis

Data collection and statistical calculations were completed in GraphPad Prism version 9.2.0 (283). Super plots were generated according to Lord et al (2020). In figure legends, n refers to individual measurements made across one trial, and N refers to the number of trials performed. Geneious Prime 2022 was used to generate the phylogenetic tree and sequence alignment (Fig S1D and E). *Chlamydomonas* sequences were generated from Phytozome 13 (Goodstein et al, 2012). To determine significance, values with $P < 0.05$ are considered significant.

## Data Availability

This study includes no data deposited in external repositories.

## Supplementary Information

## Acknowledgements

We are grateful to Chris Shoemaker and his laboratory for providing hTERT-RPE1 cells, methods, and guidance on related procedures to facilitate this work. We thank the reviewers for their careful reading, which greatly improved the study. We would also like to thank Ann Lavanway at the Dartmouth Imaging Facility for helping with immunofluorescence imaging, Radu Stan for preparing and imaging cells for electron microscopy, the BioMT Core at Dartmouth (funded by NIH NIGMS grant P20-GM113132) and Genomics and Molecular Biology Shared Resources Core at Dartmouth (funded by NCI Cancer Center Support Grant 5P30CA023108-37) for use of equipment, the Department of Anatomy and Cell Biology at the University of Kansas Medical Center where this work was started, and our funding source NIH MIRA R35GM128702 (to P Avasthi).

## Author Contributions

LL Dougherty: conceptualization, formal analysis, validation, investigation, visualization, methodology, and writing—original draft, review, and editing.
S Dutta: conceptualization, formal analysis, methodology, and writing—review and editing.
P Avasthi: conceptualization, resources, supervision, funding acquisition, project administration, and writing—review and editing.

## Conflict of Interest Statement

P Avasthi is a paid consultant for Arcadia Science.

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
