## [Reviewer comments · Life Science Alliance]

Life Science Alliance

ERK activation regulates ciliogenesis through disrupting ciliary motors, gating and microtubules

Larissa Dougherty, Soumita Dutta, and Prachee Avasthi

DOI: <https://doi.org/10.26508/lsa.202201899>

Corresponding author(s): Prachee Avasthi, Geisel School of Medicine at Dartmouth College

Review Timeline:

Submission Date:	2023-01-04
Editorial Decision:	2023-02-02
Revision Received:	2023-02-27
Editorial Decision:	2023-03-01
Revision Received:	2023-03-02
Accepted:	2023-03-03

Transaction Report:

Please note that the manuscript was reviewed at Review Commons and these reports were taken into account in the decision-making process at Life Science Alliance.

Full Revision

Manuscript number: RC-2022-01501

Corresponding author(s): Prachee Avasthi

[Please use this template only if the submitted manuscript should be considered by the affiliate journal as a full revision in response to the points raised by the reviewers.]

*If you wish to submit a preliminary revision with a revision plan, please use our "Revision Plan" template. **It is important to use the appropriate template to clearly inform the editors of your intentions.**]*

1. General Statements [optional]

This section is optional. Insert here any general statements you wish to make about the goal of the study or about the reviews.

We thank the reviewers for thoroughly examining our manuscript. We present a fully revised manuscript following our revision plan we had previously submitted. In addition to changes per reviewer suggestions, we have generated and included data from double mutants of our previously investigated map kinase phosphatases that have provided clearer and more convincing phenotypes to the BCI phenotype which has clarified this story. Because of this, we have removed data which has created some reviewer comments that no longer apply. We have addressed these below.

Among the changes we have made, we have chosen to remove single mutant data past the first figure so that the message can focus on the mutants that contain the phenotype closest to BCI, and we have removed most of the supplemental data for the microtubule section in Figure 6, again to emphasize a short clear message regarding the most important and dramatic effects of BCI on microtubules. Finally, we have removed Figure 3J and the corresponding supplemental figure regarding BCI's effect more broadly on ciliary proteins. We agree with reviewers and ASAPbio Crowd Preprint Review comments that this data was confusing and not convincing. We believe these changes have created a more concise and clear story with regards to the impact of the MAPK pathway on ciliogenesis.

This section is mandatory. Please insert a point-by-point reply describing the revisions that were already carried out and included in the transferred manuscript.

Reviewer #1 (Evidence, reproducibility and clarity (Required)):

SUMMARY:

The authors investigated the effects of an allosteric inhibitor of DUSP (BCI) on cilia length regulation in *Chlamydomonas*. Among seven conclusions summarized in Fig. 7, BCI is found to severely disrupt cilia regeneration and microtubule reorganization. Additionally, changes in kinesin-II dynamic, ciliary protein synthesis, transition zone composition and membrane trafficking are also explored. All these aspects have been shown to affect cilia length regulation. Findings from this body of work may give insights on how MAPK, a major player in cilia length regulation, functions in various avenues. Additionally, the study of BCI and other specific phosphatase inhibitors may provide a unique addition to the toolset available to uncover this important and complicated mechanism.

MAJOR COMMENTS

Major comment 1

The addition of BCI increases phosphorylated MAPK in *Chlamydomonas* based on Fig 1B. However, the claim that BCI inhibits *Chlamydomonas* MKPs is not supported at all. SF1A shows CrMKP2, 3 and 5 are related to each other but distant from HsDUSP6 and DrDUSP6. At the same time, 2 out of 3 predicted BCI interacting residues are different from the Hs and Dr DUSP6 in SF1B, contradicting "well conserved" in line 172. Consistently, mutants of these orthologs have little to no ciliary length and regeneration defects compared to BCI treatment (see major comment 6 about statistical significance). I am not convinced that BCI inhibits the identified orthologs or any MKPs in *Chlamydomonas*. It's possible that BCI inhibits a broad range of phosphatases including the ones listed and/or those for upstream kinases. But such a point is not demonstrated by the presented data.

While BCI is predicted to interact with these residues, it is also predicted to interact with the "general acid loop backbone" by fitting in between the $\alpha 7$ helix and the acid loop backbone (Molina et al., 2009). We have included the general acid loop backbone in the sequence alignment which also shows partial conservation.

MKP2 has ciliary length defects compared to wild type, though it regenerates normally. In addition, we have crossed these mutants together and have found that cells (2x3 12.2 and 3x5 29.4) cannot regenerate cilia back to normal or even mutant lengths. We have included this data in figure 1 and performed follow up analyses on these double mutants. Because these structures are not 100% conserved, and we have changed the text to "partially conserved" to reflect this, it is possible that BCI is hitting multiple DUSPs rather than just one, or the DUSPs may serve compensatory functions that rescue ciliary length.

Major comment 3

The claims that "BCI inhibits KAP-GFP protein expression" (line 271) and "BCI inhibits ciliary protein synthesis" (line 286) are not convincingly demonstrated. Overlooking that only KAP is investigated instead of kinesin-II, none of the relative intensity from the WB in 30 or 50 μM BCI and the basal body fluorescence intensity indicates a statistically significant difference. The washout made no difference in any of the assay and it's not explained how phosphatase inhibition by BCI might affect overall ciliary protein synthesis. The claims about protein expression may need a fair amount of effort and time investment to demonstrate, therefore I suggest leaving these out for this manuscript.

Though it's very interesting to see that in SF 2C cilia in 20 μM BCI treatment can regenerate slowly. Line 162, the author claimed "In the presence of (30 μM) BCI, cilia could not regenerate at all (Fig 1E)". Since Fig 1E only extends to 2 hours, I think it's important to clarify if in 30 μM BCI cilia indeed can not generate even after 6 or 8 hours.

We have altered the text to be more specific with our wording that KAP-GFP is investigated rather than kinesin-2, and we have added text to indicate that downstream phosphorylation events could impact transcription and translation of proteins necessary for ciliary maintenance. This

interpretation of the data mentioned above is correct; KAP-GFP is not significantly altered at the basal bodies or in accordance with the steady state western blots. What we see here and demonstrated in Figure 2F-I is the depleted KAP-GFP protein which is not restored following a 2 hour regeneration in BCI. We likely do not see a difference in steady state conditions because the protein is not degraded, just being moved around in the cell. We can only see the difference when the majority of KAP-GFP, which the data suggests is mostly present in cilia, is physically removed through ciliary shedding. This protein is not replaced during a 2 hour regeneration which allows us to conclude that this protein is inhibited due to BCI.

~~The washout made a small difference in the double regeneration whereby we begin to see cilia begin to form in washed out conditions, though this was not statistically significant. It is possible that BCI has a potent effect on the cell similar to how other drugs, such as colchicine, cannot be easily washed out. The purpose here is to show that regardless of the statistical significance, cells can begin to regenerate their cilia after BCI washout, though this occurs 4 hours after washout in doubly regenerated cells, and we do not see this potent effect on the singly regenerated cells in SF 2C. Though in SF2C, as mentioned, we do see slowly growing cilia, and this could, once again, be due to the potent inhibition BCI has on ciliary protein synthesis. We will confirm and clarify if 30 μ M BCI cannot regenerate even after 6 or 8 hours.~~ **Based on this comment, comment from reviewer 2, and ASAPbio Crowd Reviews, we agree that the conclusion for total ciliary protein synthesis is confusing and not convincingly demonstrated, so we have decided to remove this data from the manuscript.**

Major comment 4

A single panel in Fig 4A also can't support the shift in protein density in the TZ in line 317. As line 324 implies protein synthesis defect by BCI, the very minor (in amount and significance) reduction of the NPHP4 fluorescence should not be interpreted as any disruption at all to the transition zone. I suggest checking other TZ proteins such as CEP290 etc or leave this section out.

Also, The additive effect from BFA and BCI treatment in Fig 5A suggests BCI affects cilia length independent of Golgi. The "actin puncta" and arpc4 mutant are not sufficiently introduced. And more importantly, how increase in the actin puncta explains the shorter cilia length caused by BCI while actin puncta are absent in arpc4 mutant with shorter cilia? Also, the Arl6 fluorescence signal "increase" is not significant in either time point. I suggest leaving this section out as well.

We agree that one EM image cannot support a protein shift and have removed our observation in the text. However, we do see a statistically significant decrease in NPHP4 fluorescence in BCI treated cells which we consider a disruption in the sense that the structural composition is altered. We will change the word "disruption" to "alteration" for clarity. Though this is a minor defect, we believe it is still worth noting. We believe this data still adds to the model that though the EM-visible structure is unaltered, finer details within the transition zone are indeed altered and we cannot rule out that these smaller changes are not impacting protein entry into cilia. Awata et al. 2014 shows that NPHP4 is important for controlling trafficking of ciliary proteins at the transition zone, and its loss from the transition zone has been found to have effects in ciliary protein composition. Because we see decreased NPHP4 expression, we believe this is a notable finding as we see effects on the abundance of a protein which is known to affect ciliary protein composition and have therefore chosen to leave the data in the manuscript. **We have adjusted the language to most accurately describe our findings.**

We also agree with the interpretation that the additive effect seen from BFA and BCI treatment could suggest independent pathway collapse separate from the Golgi which we have mentioned in the manuscript.

We have provided more information to introduce actin puncta and ARPC4 with regards to membrane trafficking. Bigge et al. 2020 shows that ARPC4, a subunit of the ARP2/3 complex which is an actin binding protein important for nucleating actin branches, has a role in ciliary assembly.

ARPC4 mutants have repressed ability to regenerate their cilia. One feature they noticed in regenerating cells is the immediate formation of actin puncta which are reminiscent of yeast endocytic pits. This observation in addition to altered membrane uptake pathways in *Chlamydomonas* suggests that ciliogenesis involves reclaiming plasma membrane for use in ciliogenesis (because of the diffusion barrier preventing a contiguous membrane). Here, we incorporate this assay to assess the ability for the cell to reclaim membrane during BCI treatment and find that there is increased actin puncta. This could indicate that there is increased number of endocytic pits or alternatively that the lifetime of these pits is increased (perhaps due to incomplete endocytosis) such that we are able to detect more of them at a fixed point in time. While we cannot say which is happening here, we have previously found that these actin puncta are likely endocytic and needed to reclaim membrane for early ciliogenesis. An increase in these puncta may suggest dysregulated endocytosis in one way or another. ARPC4 cells cannot form the actin puncta in the first place, whereas we are seeing defects following puncta formation. We have taken out the Arl6 data.

Major comment 5

It is very interesting that BCI disrupts microtubule reorganization induced by deciliation and colchicine. Data in Fig 6B and C are presented differently than those in SF 4C. For example, in SF 4C, BCI treatment for 60 min has close to 50 % cells with microtubule partially reorganized while in Fig 6C about 20% cells with microtubule fully (or combined?) reorganized. The nature of the difference is unclear to me without an assay comparing the two directly. Hence the implied claim that BCI affects colchicine induced microtubule reorganization differently than deciliation induced one is hard to interpret (line 398, line 388 vs line 403).

The fact that taxol doesn't rescue cilia regeneration defect by BCI is very interesting. Here taxol treatment results in fully regenerated cilia while Junmin Pan's group (Wang et. al., 2013) reported much shorter regenerated cilia. It might be worthwhile to compare the experimental variance as this is a key data point in both instances. The relationship between cilia regeneration and microtubule dynamic is not in one direction. On one side, there's a significant upregulation of tubulin after deciliation. While many microtubule depolymerization factors such as katanin, kinesin-13 positively regulate cilia assembly (though not without exceptions). It is hard to determine that the BCI induced cilia regeneration defect can't be rescued by other forms of microtubule stabilization. Microtubule reorganization is one of the most striking defects related to BCI treatment. I suggest changing the oversimplified claim to a more limited one (such as "PTX stabilized microtubule ...") and an expansion on the discussion about microtubule dynamics and cilia length regulation beyond the use of taxol. Meanwhile, I strongly encourage authors to continue to investigate this aspect and its connection to the cilia regeneration.

We have removed data regarding "partially" formed cytoplasmic microtubules and only included formed microtubules for each of these experiments for clarity. In addition, we have removed the original supplement 4 investigating drug interactions on microtubule stability for more focus on the role of paclitaxel vs. BCI on microtubule formation. It is important to note the different taxol concentration used here. While Wang et al., 2013 used 40 μ M taxol to study ciliary affects, we use 15 μ M where stabilization still occurs. There have been reports of varied cell responses to higher vs. lower doses of taxol (see Ikui et al., 2005, Pushkarev 2009, Yeung 1999) mostly with regards to the cell's mitotic/apoptotic response. We could be seeing altered responses at this lower concentration because *Chlamydomonas* cells also behave differently in higher vs. lower taxol concentrations. Thank you for your suggestions. We have adjusted the text to be more specific to PTX treatment as opposed to general stabilization.

Major comment 6

Throughout this manuscript, the standard the authors used to interpret statistical significance is erratic. In a few instances, the threshold for p value is clearly indicated such as in Fig 1 legend. Though other times, much higher p values are considered differences.

Here are some examples:

SF 1C, $p=0.1167$ is considered "(mkp5) shorter than wildtype ciliary lengths" (also line 177 "SF 1C" instead of "SF 1D")

Fig 3C, $p=0.083$ interpreted as "slightly less" in line 262 and possibly as "(KAP-GFP) not being able to enter (cilia)" in line 268

Fig 3G, $p=0.1087$ is considered "not decrease after two hours" line 267

SF 3C, $p=0.2929$ for mkp2 mutant (misuse of "orthologs" in line 352) is considered "fewer actin puncta compared to wild type cells" (line 352).

SF 6B, $p=0.1565$ for mkp3 mutant (line 421: misuse of "orthologs" and correct use of "ortholog mutants") is considered not be able to "fully reorganize their microtubules" (line 421).

These instances sometimes serve as basis for major conclusions and should be clarified or more carefully characterized.

We agree the interpretations are very erratic in places and greatly appreciate this detailed list making it easy to find and correct these interpretations. We have adjusted the text in the mentioned places to reflect these changes, and we have made a statement in the text and under statistical methods that say we consider $p < 0.05$ significant. *We have made figure 1 and other instances consistent by providing the numeric p values as well.*

Major comment 7:

There are several places where the technical detail or presentation of the data are missing or clearly erroneous.

Fig 1B: pMAPK and MAPK antibodies used in the WB are not described in the Material and methods. It's not clear if the same #9101, CST antibody used for RPE1 cell in Fig 1J is used.

We have updated the materials and methods to include that this antibody was used for both RPE1 and *Chlamydomonas* cells.

line 260 and Fig 3A state 20 μM BCI was used while Fig 3 legend repeatedly states 30 μM until (J). Also 30 μM in SF 2A.

We have corrected the text to 20 μM BCI in the mentioned places.

Fig 6C, the two lines under p value on top mostly likely start from the second column (B) instead of the first (D). Fig 6G, the line is perhaps intended for the second and fourth columns?

We have added brackets to make these comparisons clearer. We had performed a chi-square analysis and were comparing the difference between DMSO and BCI before PTX stabilization or MG132 treatment to after.

Fig 6C, legends indicate bars representing each category. But only one bar is shown for each column. Same for 6G?

This is the same as the previous comment for the way we represented the statistics. *We have added brackets to show the comparisons.*

Minor comments:

1. A number of small errors in text were noted above.

Done.

2. "orthologs" is misused in place of "ortholog mutants": line 176, 352, 421 (first), 879, 882, 898, 902, 938, 939.

Done.

3. Capital names is misused as mutant names (e.g. "MKP2" should be "mkp2"): line 178, SF 1C, 1D and 1E, SF 3C, SF 6A

Done.

4. At several places such statistical analysis lines indicated are chosen confusingly. A simplest example is in Fig 1D, the comparison between 0 to 45 is less important than 0 to 30. Same as in Fig 1H, 1I. The line ends are inconsistent as well. They either end in the middle or the edge of the columns/data points (such as in SF 4B) and some with vertical lines (SF 2B, SF 4A, SF 6B). I suggest adding vertical lines pointing to the middle to indicate the compared datasets clearly.

Thank you for this suggestion. We agree and have updated the figures with brackets to clear up this confusion.

5. line 101 remove "the"

Done.

6. line 120 "modulate" to "alter"

Done.

7. line 198 "N=30" should be "N=3"

Done.

8. line 212. The legend for p value is likely for (G)

Done.

9. line 284, "singly" should be "single"

Done.

10. The dataset for "Pre" and "0m" in Fig 6D and 6E are clearly the same. Consider combining the two as in Fig 6C.

Done.

11. Fig 6E, "BCI" on the X-axis should be "DMSO".

Done.

12. line 685, remove "?".

Done.

13. line 894: "Fig 3J" instead of "Fig 3H"

Done.

14. SF 1 legend, (C) and (D) are inverted.

Done.

15. SF 4A "Recovered" should be "Full"

Done.

16. SF 5, row 5, under second arrow perhaps missing +PTX

Done. We greatly appreciate this close reading of the text and the list of changes making these errors easy to find. We have made these changes in the manuscript.

Reviewer #1 (Significance (Required)):

Increasing evidence indicates that several MAPKs activated by phosphorylation negatively control cilia length while few studies focus on how MAPK dephosphorylation affects cilia length regulation, largely due to the unknown identity of the phosphatase(s) specifically involved in cilia length regulation. The authors set out to investigate the effect of BCI on cilia length control. BCI specifically inhibits DUSP1 and DUSP6, both of which are known MAPK

phosphatase, and therefore may provide a unique opportunity to understand how MAPK pathway is controlled by specific phosphatase(s) activity in cilia length regulation.

Overlooking some inconclusive results and oversimplified interpretations, I find the most striking findings are the BCI's effects including ciliogenesis, kinesin-2 ciliary dynamics and microtubule reorganization. I believe these findings have significant relevance to the stated goal (line 131) and conclusions (line 57) and readers may find them a good starting point for further investigation of the role phosphatases play in cilia length regulation.

Cilia length regulation is a complicated mechanism that is affected by many aspects of the cell and functions differently in various systems. My field of expertise may be summarized by cilia biology, cilia length regulation, IFT, kinesin, kinases (MAPKs), microtubules. The membrane trafficking's role in cilia length regulation is somewhat unfamiliar to me. Additionally, the authors used a number of statistical tests and corrections in various assays. The nuance of these choices is not clear to me and neither explained to general readers.

Reviewer #2 (Evidence, reproducibility and clarity (Required)):

In their manuscript, "ERK pathway activation inhibits ciliogenesis and causes defects in motor behavior, ciliary gating, and cytoskeletal rearrangement," Dougherty et al investigate how BCI, an activator of MAPK signaling, regulates ciliary length. Despite advances in our understanding of the structure and function of cilia, a fundamental question remains as to what are the mechanisms that control ciliary length. This is a critical question because cilia undergo dynamic changes in structure during the cell cycle where they must disassemble as they enter the cell cycle and must rebuild after cell division. This work contributes to a growing body of work to determine mechanisms that regulate cilia length.

The authors use a well-established model system, *Chlamydomonas*, to study cilia dynamics. This work expands on previous findings from these authors that inhibition of MAPK signaling using U0126 lengthens cilia as well as other publications that implicate MAPK signaling in controlling ciliary length. However, the authors only observe a few significant phenotypes with other subtle trends, leaving the conclusion regarding the role of MAPK signaling murky. Furthermore, it is unclear through what mechanism BCI impacts ciliary length. Several issues must be addressed:

MAJOR ISSUES

1. The basis for this study is the use of the ERK activator BCI, which the authors show activates MAPK signaling. While the authors do use putative DUSP6 ortholog mutants to corroborate some of the phenotypes, the majority of the data (and conclusions) uses BCI. However, there may be off target effects and the authors do not address this limitation of the study. The authors only use 1 pharmacological tool to manipulate MAPK signaling, so it is unclear whether these ciliary disruptions are specifically due to increased MAPK. It is necessary to clarify the following questions about BCI action to interpret the results:

- **a.** What are off target effects of BCI? Does BCI impact proliferation? Why is the BCI phenotype of cilia shortening transient and dose dependent? Why does the phenotype of cilia length and regeneration capacity in *Chlamydomonas* differ from both ortholog mutants and hTERT-RPE1 cells?

While we do mention following supplemental figure 1 that other MKPs could be the target for BCI, we also cite Molina et al., 2009 who showed specificity for BCI hydrochloride in zebrafish. BCI targets primarily DUSP6, but also exhibited some activity towards DUSP1. In this study, the authors had also used zebrafish embryos to check expression of 2 other FGF inhibitors, spry 4 and XFD, in the presence of BCI but found that their effects were not reversed. In addition, they checked the

ability for BCI to suppress activity of other phosphatases including Cdc25B, PTP1B, or DUSP3/VHR and found that BCI could not suppress these phosphatases. BCI inhibition has previously been found to be more specific to MAPK phosphatases. In addition, we have previously confirmed that U0126 has a slight lengthening effect on *Chlamydomonas* which further implicates this pathway in cilium length tuning (Avasthi et al. 2012).

While cell proliferation assays maybe provide more support for MAPK signaling, it does not clarify lack of off target effects that could also contribute to this same phenotype. We had previously provided a cell proliferation assay for RPE1 cells where we show that higher concentrations of BCI result in cellular senescence as well (Fig 11), though we have since removed this data and the 48 hour ciliary growth data to emphasize the more robust BCI phenotype occurring during the 2 hour time scale which we use throughout the rest of the paper.

The BCI phenotype of cilia shortening is likely transient and dose dependent due to its effect on ciliary protein synthesis demonstrated in the previous Figure 3J. The increase in drug likely increases its substrate binding to exert its effects on the cell faster, even if this includes off target proteins.

In RPE1 cells, we are likely seeing differences in regeneration capacity potentially due to their different mechanisms of ciliogenesis (RPE1 cells partake in intracellular ciliogenesis where axonemal assembly begins in the cytosol whereas *Chlamydomonas* cells partake in extracellular ciliogenesis where axonemal assembly begins after basal bodies dock to the apical membrane), or it could be that we're missing a delay in regeneration in RPE1 cells after waiting 48 hours for ciliogenesis. We do not check this process sooner. There may be a defect that cells overcome. Additionally, among ortholog mutants and RPE1 compared to BCI-treated wild-type *Chlamydomonas*, there indeed could be off target effects or the drug could be targeting all of these MKPs rather than just one. We will add this to the discussion for clarity.

Reviewer #2 (Significance (Required)):

2. In multiple instances the conclusions are overstated, and the author must clarify the interpretation of the results to reflect the data presented. Here are some examples:

- **a. The conclusion that protein synthesis is disrupted is incorrect in two instances (line 258 and 275) as the experiments in figure 3 do not directly examine changes in synthesis (they look at cilia regeneration as a proxy).**

We show that KAP-GFP expression is not normal during regeneration at 120 minutes which suggests, in addition to the inability for cilia to grow in BCI, that synthesis is inhibited because this protein is not replaced. In addition, blocking the proteasome did not rescue this decrease in KAP-GFP expression indicating that this is not a matter of KAP-GFP protein being degraded rapidly. We use regeneration and KAP-GFP readout as a proxy for protein synthesis. We have clarified this in the text.

- **b. The conclusion that BCI disrupts membrane trafficking is too broad when the authors only examined trafficking of one membrane protein, Arl6.**

While we only looked at one membrane protein specifically, we assess other membrane trafficking paths. We looked at BCI vs. BFA to assess Golgi trafficking (Dentler 2010) in addition to formation of actin puncta which is used in Bigge et al. 2020 as an assay for membrane uptake from the plasma membrane for incorporation into cilia.

- **c. The conclusion that the transition zone is disrupted is too broad based on a decrease in the expression of one transition zone protein, NPHP4.**

We have changed the text to be more specific to NPHP4.

3. Highlighting the overstatement, the conclusion of the header and figure caption on page 10 contradict one another. The manuscript states that "BCI partially disrupts the transition zone" (line 313) and that "The TZ structure is structurally unaltered with BCI treatment" (line 329).

In the manuscript, we show that the EM-visible structure is indeed unaltered. Because we see a decrease in NPHP4 fluorescence, we concluded that while the EM-visible structure is unaltered, protein composition within the transition zone is altered which suggests that BCI partially disrupts the transition zone.

5. Why is kinesin-2 the only target studied for ciliogenesis? Ciliogenesis is a complex process that involves many other critical proteins and investigating kinesin-2 alone is not sufficient to conclude why BCI prevents cilia assembly.

We use kinesin-2 because it is the only ciliary anterograde motor in *Chlamydomonas* which is required for proper ciliogenesis. By assessing kinesin-2, we were able to address whether this protein alone was the cause for inhibited ciliary assembly (and we find that it's not), whether its ability to enter was impacted (likely owing to defects in other protein entry), and we were able to use this protein to understand how its protein expression was affected. Because KAP-GFP is a cargo adaptor protein and interacts with IFT complexes and other cargoes, defects in this protein can have a wide range of implications. We agree and the data agree that kinesin-2 alone is not sufficient to conclude why BCI prevents cilia assembly. Because of this, we assessed other pathways including membrane trafficking and microtubule stabilization to better understand why we see defects in ciliary assembly. Certainly many other proteins are important in ciliogenesis and we hope that this study sparks further work in this area to identify additional causative explanations for impaired ciliogenesis upon MAPK activation..

6. Tagged ciliary proteins are sensitive to disruptions in function and expression within cilia. It is important to include proper controls in the study using KAP-GFP *Chlamydomonas* cells to ensure that KAP-GFP maintains endogenous expression levels and normal function as untagged KAP. Furthermore, if this information is available through the resource where the cells were purchased, then this needs to be discussed.

KAP-GFP expressing *Chlamydomonas* has previously been validated as described in Mueller et al., 2005. We have included this detail in the text.

7. The authors need to provide clear explanations to a general audience of why this technique is used and how the authors reached the interpretations. There are several instances where the authors use techniques that are cited as fundamental papers in *Chlamydomonas*. Here are two examples:

- **a. It is unclear how the authors concluded that decreased frequency and velocity of train size shows that kinesin entry, specifically, is disrupted.**

We have expanded on this in the text. Please see response to reviewer 1, Major comment 2 above.

- **b. It was impossible to follow how the experiment where cells treated with cycloheximide could not regenerate their cilia following BCI treatment shows that BCI inhibits protein synthesis.**

Because of the confusing conclusions surrounding this data set, we have chosen to omit this data from the most recent version of the manuscript. In this experiment, we deplete the ciliary protein pool by forcing ciliary shedding two times. Following the first shedding, there is enough protein to assemble cilia to half length (Rosenbaum, 1969). We ensure that the protein pool is completely used up by inhibiting further ciliary protein synthesis with cycloheximide. For the second shedding event, completely new ciliary protein must be synthesized for ciliogenesis to occur which is why ciliogenesis takes much longer compared to a single regeneration where half of the ciliary protein pool still remains and can be immediately incorporated into cilia (SF 2C – also removed in the new version).

In the presence of BCI, cilia cannot grow at all as expected; but 4 hours after BCI is washed out, we see ciliogenesis just beginning to occur which indicates that there is protein present for ciliogenesis to begin whereas in cells where BCI is not washed out, we do not see any ciliogenesis.

8. The impact of BCI treatment on membrane trafficking as presented is confusing. BCI exacerbated the effects of BFA treatment on Golgi, yet the authors do not address that this could be an indirect effect of BCI or an off-target effect of BCI.

This is addressed in the discussion (paragraph 4).

9. The discussion section includes many interpretations of the results, but leaves the reader confused as to what the authors think might be happening. The manuscript would be far clearer if the authors would provide a working model for why BCI impacts cilia length. It is fine for this to be left for future work but, as the experts, the authors must have relevant thoughts to share with the field.

Figure 7 provides a model with as much as we can conclude given the data; what we show is that BCI inhibits many different processes in the cell, but we do not necessarily show links between these processes to provide a complete working model of how these are all interconnected; we have provided a summary model that depicts the various, still disconnected processes that are inhibited by BCI. MAP kinases such as ERK have dozens of downstream targets both within and outside the nucleus. Ciliogenesis also is a complex process coordinating many cellular mechanisms. The intersection of these two seem to have a multi-fold effect that results in a dramatic ciliary phenotype through a combination of factors, however not one that fully explains the severity upon initial deciliation in BCI/MAPK activation. Further work is needed to identify the precise cause of completely inhibited cilium growth from zero length.

MINOR ISSUES

1. The title of the manuscript is inaccurate and overstates the pathway involvement in cilia. The authors do not directly show that ERK pathway activation causes the ciliary phenotypes due to the use of BCI, a drug that modulates ERK.

We have adjusted the title to “The ERK activator, BCI, causes...”

2. When discussing results of data that are not statistically significant it creates confusion to state that the results "increased/decreased slightly".

We agree that references to statistics are inconsistent or confusing throughout the text and have adjusted these references accordingly.

Reviewer #3 (Evidence, reproducibility and clarity (Required)):

SUMMARY:

In this study, the authors used a pharmacological approach to explore the function of ERK pathway in ciliogenesis. It has been reported that the alteration of FGF signaling causes abnormal ciliogenesis in several animal models including *Xenopus*, zebrafish, and mice. However, it remains elusive the molecular detail of how ERK pathway is associated with cilia assembling process. The authors found that the ERK1/2 activator/DUSP6 inhibitor, BCI inhibits ciliogenesis, highlighting the importance of ERK during ciliogenesis. Overall, this paper is well written, data are solid and convincing. This paper will be of great interest to many researchers who are interested in understanding ciliogenesis. The following comment is not mandatory requests but suggestions to improve the paper's significance and impact.

MAJOR COMMENTS:

- Combination of chemical blocker experiments were well controlled and data are solid. The authors are aware of the side effects of BCI, thus they carefully characterized the phenotypes of Mkp2/3/5 in *Chlamydomonas*. This reviewer wonders if the levels of ERK1/2 phosphorylation are activated in these mutants. Did the authors examine the levels of ERK1/2 phosphorylation in these mutants?

While we did not originally include the data showing ERK activation in these mutants, we have checked pMAPK activation and found that it is not significantly upregulated in these mutants. This could likely be due to compensatory pathways preventing persistent pMAPK activation. For example, constant ERK activation can lead to negative feedback to regulate this signal for cell cycle progression (Fritsche-Guenther et al., 2011). The ERK pathway has not been fully elucidated in *Chlamydomonas*, but it is possible that these similar mechanisms are in place for MAPKs. We have included our analysis of BCI-induced pMAPK expression in figure 1.

Reviewer #3 (Significance (Required)):

Accumulated studies suggest that the FGF signaling pathway plays a pivotal role in ciliogenesis. Disruption of either FGF ligands or its FGF receptor results in defective ciliogenesis in *Xenopus* and zebrafish. On the other hand, FGF signaling negatively controls the length of cilia in chondrocytes that would cause skeletal dysplasias seen in achondroplasia. Therefore, there is strong evidence suggesting that FGF signaling participates in ciliogenesis in cell-type and tissue-context dependent manners. However, the detailed mechanism of the downstream of FGF signaling in ciliogenesis is still unclear. In this regard, this paper is beneficial for the cilia community to expand the knowledge of how ERK1/2 kinase contributes to the regulation of ciliogenesis.

This reviewer therefore suggests that the authors may want to add more discussion to explain how their finding possibly moves the field forward to understand the pathogenesis of multiple ciliopathies.

We have added the following text to the discussion:

“Further work is needed to identify the precise cause of completely inhibited cilium growth from zero length. Regardless, the insight provided here can help to provide a basis for understanding more broad mechanisms for how ciliopathies or manipulated cilia can drastically alter the fate of the cell.”

Major comment:

- If the authors want to emphasize their finding is associated with MAP kinases, it would be also beneficial to examine other major MAP kinase pathways such as P38/JNK. If not, then this reviewer suggests revising the text as ERK through this manuscript to avoid confusions. Because the ERK pathway has not been fully elucidated in *Chlamydomonas*, we have refrained from using “ERK” as a descriptor because this particular MAPK shares equal identity with multiple MAPKs in *Chlamydomonas*. Further, BCI may be targeting more than one MAPK phosphatase resulting in the myriad phenotypes we have discovered. At this time, we lack a level of gene-level resolution to map to known MAPK pathways.

MINOR COMMENTS:

- I suggest moving supplemental figure 1 to the main figure (Fig. 1?) so that the readers appreciate the author's careful examination of BCI through this manuscript.

We have moved the mutant data from the supplement to the main figure 1.

1. Description of analyses that authors prefer not to carry out

Please include a point-by-point response explaining why some of the requested data or additional analyses might not be necessary or cannot be provided within the scope of a revision. This can be due to time or resource limitations or in case of disagreement about the necessity of such additional data given the scope of the study. Please leave empty if not applicable.

Reviewer 1:

Major comment 2

The claim that "BCI treatment decreases kinesin-2 entry into cilia" (line 236) is a misinterpretation of the data presented. The data indicates KAP-GFP have reduced accumulation in cilia, decreased IFT (anterograde) frequency, velocity and injection size associated with BCI treatment. Though as shown in Fig 1D and Fig 2C, cilia length is also shorter due to BCI treatment. Ludington et. al, 2013 showed a negative correlation of cilia length and KAP injection rate in various treatments that affect cilia length. It's essential to rule out that the KAP dynamics reported in the current manuscript is not an outcome of shortened cilia in order to claim as line 236 seems to suggest. One way to demonstrate specific effect by BCI would be to compare KAP dynamic in cilia with equal or similar length, either by only selecting the shorter cilia from wt or use other treatments that are known to decrease cilia length (chemicals, cell cycle, mutants etc.). Given the capability and resource represented in this manuscript, I don't expect a significant cost and time investment for these experiments.

Ludington et al., 2013 shows that injection size decreases with increasing length. Our data show that the shorter length cilia have decreased injection size and rate inconsistent with the cause being due to shortened length alone. In other words, in figure 2C and 2G, we see decreased KAP-GFP fluorescence in shorter cilia as opposed to greater fluorescent signal in shorter cilia seen in Ludington et al., 2013. This data, in combination with the decreasing frequency of KAP-GFP entry overtime in figure 2E and decreased velocity in figure 2F support decreased kinesin-2 entry into cilia. If entry was unaltered, we would expect increased KAP-GFP fluorescence in the cilia over time in BCI-treated cells.

Reviewer 2:

4. The authors state that the decreased length of cilia following BCI treatment could be a result of reduced assembly or increased assembly. Disruptions to cilia assembly and disassembly are not mutually exclusive and both must be evaluated. The authors do not test whether cilia disassembly is disrupted in BCI treatment and therefore, cannot conclude that BCI solely disrupts cilia assembly.

While effects on disassembly remains a possibility, the striking inability to increase from zero length upon deciliation and the effects on anterograde IFT through the TIRFM assays suggest an affect on assembly. There may be effects on disassembly and likely many other cilia related processes not investigated but we feel it remains accurate to conclude that assembly is affected by BCI treatment.

Reviewer 3:

- If time allows, in addition to examining NPHP4, it would be beneficial to examine other TZ/TF markers such as CEP164 to confirm if BCI partially disrupts the TZ.

Given the known outcomes of NPHP4 loss in *Chlamydomonas* (Awata et al., ...) in affecting ciliary protein composition, we suspect the changes in NPHP4 abundance at the transition zone will have a significant impact and agree it would be interesting in a follow up study to see how other transition

Full Revision

zone proteins (particularly ones known to interact with NPHP4 or others critical for TZ function) are impacted following BCI treatment.

February 2, 2023

Re: Life Science Alliance manuscript #LSA-2023-01899

Prachee Avasthi
Geisel School of Medicine at Dartmouth College

Dear Dr. Avasthi,

Thank you for submitting your revised manuscript entitled "ERK pathway activation inhibits ciliogenesis and causes defects in motor behavior, ciliary gating, and cytoskeletal rearrangement" to Life Science Alliance. The manuscript has been seen by the original reviewers whose comments are appended below. While the reviewers continue to be overall positive about the work in terms of its suitability for Life Science Alliance, some important issues remain.

Our general policy is that papers are considered through only one revision cycle; however, given that the suggested changes are relatively minor, we are open to one additional short round of revision. Please note that I will expect to make a final decision without additional reviewer input upon resubmission.

Please note that new experimental data is not expected in response to Reviewer 1's point #2, unless it is readily available.

Please submit the final revision within one month, along with a letter that includes a point by point response to the remaining reviewer comments.

To upload the revised version of your manuscript, please log in to your account: <https://lsa.msubmit.net/cgi-bin/main.plex>
You will be guided to complete the submission of your revised manuscript and to fill in all necessary information.

B. MANUSCRIPT ORGANIZATION AND FORMATTING:

Sincerely,

Reviewer #1 (Comments to the Authors (Required)):

The authors made significant changes and additions in this revision compared to the previous submission for which I served as reviewer #1 for Review Commons. As the authors summarized, this revision strengthened their interpretation and conclusion in

several aspects, particularly the clarification of statistical significance and the assays involving double mutants. Additionally, the removal of several confusing or hard to interpret data sets allow the main observations to be the focus of this manuscript. I agree with the authors that this revision presents a much clearer and more concise story overall.

This revision continues to make seven major claims summarized in Figure 7. Based on the updated manuscript, below are my assessments on each main point of the paper.

1. BCI induced MAPK phosphorylation disrupts ciliogenesis.

While the single mutants do not show significant defect in ciliogenesis following deciliation by pH shock, two of the three double mutants partially phenocopy the BCI induced ciliogenesis phenotype. Consistently, the change in MAPK photophosphorylation was also most prominent in these two mutants. Given the background and scope of this manuscript, I would consider the interpretation supported with limitations.

2. BCI disrupts KAP-GFP protein synthesis during ciliogenesis and ciliary dynamics.

The reported changes in Fig. 2 are consistent with a disruption in KAP-GFP dynamics.

For reduced KAP-GFP expression, statistical significance directly comparing the third and fourth column in Fig 3C (similar to in 3D) is needed. A secondary, preferably a more sensitive method should be employed: for example, while the basal body fluorescence did not change significantly, did the overall fluorescence or the part in the cell body except basal bodies change? Or is there a major change in transcription/mRNA level? These might take several weeks to test depending on the technical limitations.

3. BCI disrupts NPHP4 expression and turnover at the transition zone.

The decrease in NPHP4 expression reported here is categorically different from the *nphp4* mutant (null) reported in Awata et al. 2014 and is marginally significant based on the authors' criteria. Additionally, the authors pointed out BCI "does not grossly disrupt the transition zone as a mechanism for altering ciliogenesis" (line 309). Therefore, I do not think the evidence provided provides satisfactory support to this point. The author may demonstrate a dosage-dependent impact of NPHP4 at the level demonstrated in this dataset or directly test that NPHP4 turnover is indeed altered due to BCI. Given these may take considerable effort and time, I would like to reiterate my previous comment (Major comment #4) about leaving this part out.

4. BCI disrupts membrane trafficking.

The increase in actin puncta in the *mkp2 x mkp3* double mutant is particularly shocking given that all three single mutants showed reduction in actin puncta in previous submission (previous Sup Fig 3C). Similarly, the *mkp3 x mkp5* double mutant rescued both single mutants (again, from previous Sup Fig 3C). Can the authors provide any explanation or interpretation of such a difference?

The updated introduction about actin punta and ARPC4 are very helpful in orienting the interpretation. As long as the authors can provide satisfactory explanation about the discrepancy (?) above, the interpretation of this data is valid.

5. BCI disrupts cytoplasmic microtubule reorganization.

6. PTX stabilized microtubules do not rescue BCI induced ciliogenesis defects.

These two parts were and are the highlight of this manuscript. The changes in this revision materially improved this section and made the interpretation much easier to comprehend. One point that needs clarification is what the authors mean by the double mutants "could not fully reorganize their microtubules", as the changes from pre to 60m in SF 4B are not different between WT and double mutants.

Minor comments:

Line 238, unsupported claim: unclear if the decrease in fluorescence is either ciliary length depend or significant

Line 299, mis interpretation: no decrease of KAP-GFP at the basal bodies, see line line 277 and Fig 3D.

Line 310, unsupported claim: protein turnover untested.

Line 331, unsupported claim (yet): at this point, the additive effect of BCI and BFA only exclude Golgi from potential membrane trafficking related targets of BCI.

Line 453, misinterpretation: the data showed MTs organization defect induced by BCI does not contribute to the early ciliogenesis phenotype. The conclusion can't be broadened since it's only tested by rescuing with PTX.

Larisee Dougherty et al. explored how MAP kinases can regulate ciliary length using an unusual approach, focusing on a phosphatase inhibitor, BCI, that activates the MAP kinase. This study represents a new way of researching this topic and the provided data is of great value to the community. Given satisfactory response to the comments above, I consider this manuscript a strong fit for Life Science Alliance.

Reviewer #2 (Comments to the Authors (Required)):

The revised manuscript is greatly improved, and the authors successfully addressed many of the reviewer comments. The authors provided a complete and convincing rebuttal, however, some of this logic and language is not present in the final manuscript. To effectively convince the readers of the results, the authors should incorporate more points from their rebuttal in the main text. The authors did this in several instances. Two examples include expansion of the actin puncta and KAP-GFP dynamics experiments/analysis. Here we outline the other instances where additional text or changes to the summary figure are necessary to clarify the results and interpretations.

MAJOR ISSUES

1. We appreciate that the manuscript includes references examining the off-target effects of BCI in zebrafish. This does not

- entirely address the possibility of Chlamydomonas specific off target effects of BCI. Please include language regarding how off target effects of BCI have not been directly explored in Chlamydomonas and to comment on why BCI was chosen over other pharmacological tools.
2. It is possible that BCI could perturb the cell cycle, and therefore indirectly affect cilia assembly and disassembly. The authors begin to address this in the discussion, "BCI could arrest cells at the interphase checkpoint or induce cell cycle arrest which could... shift to assembly or disassembly". Please expand on this limitation in the discussion.
 3. In response to our question "why is the cilia length and regeneration phenotype different in Chlamydomonas compared to RPE1 cells" the authors provide a clear response in the rebuttal. Please include this language in the manuscript with the hTERT-RPE1 results.
 4. To address our critique that protein synthesis is not directly examined, the authors state in their rebuttal that they "use KAP-GFP readout as a proxy for protein synthesis". This language must be included in the manuscript to clarify the interpretation of the results in Fig. 3. While there is some evidence that protein degradation is not impacted (an n=1 is incomplete), we ask the authors to moderate the conclusion that protein synthesis is decreased throughout the manuscript. This result does not eliminate alternative possibilities, and this should be addressed in the discussion.
 5. Disruptions to cilia assembly and disassembly are not mutually exclusive. The text needs expanding to include the possibility that BCI could impact disassembly, as disassembly was not tested.
 6. We appreciate that the authors changed the language around Kinesin-2 to reflect the direct observations of KAP-GFP. The authors also state in the rebuttal that "Because KAP-GFP is a cargo adaptor protein and interacts with IFT complexes and other cargoes, defects in this protein can have a wide range of implications". Please include this text in the manuscript to expand on the interpretation of Fig. 3.
 7. The authors do a great job addressing individual points in the rebuttal, but this not reflected in the discussion section. The discussion still falls short on synthesizing the new results. As written, the discussion summarizes all the results as independent processes. Instead, the discussion should address how these disrupted processes all connect to ciliogenesis, which is the predominant phenotype following BCI treatment. Similarly, the final figure should reflect how the results all connect to ciliogenesis instead of disparate consequences of BCI treatment. Currently, the boxes in the figure overstate the results (ex: transition zone protein composition is an overstatement of NPHP4 protein expression) and expanded language in the figure legend or directly in the figure is necessary to reduce these broad conclusions.

MINOR ISSUES

1. While the experiment with cycloheximide was removed, the cycloheximide text is still present in the discussion, methods, and references.
2. Line 299 states "Decreased KAP-GFP expression at the basal bodies..." which contradicts the result in Fig. 3D that there is no change in KAP-GFP expression at the basal bodies.
3. Line 296 should write "Given the effect of BCI on KAP-GFP ciliary entry,..."
4. Remove "and hTERT-RPE1" from the Figure 1 legend since those results are now in supplemental.

We greatly thank the reviewers for their close and thorough examination of the manuscript. We have revised the manuscript accordingly and include responses to reviewers below. We have fixed the noted inconsistencies and unsupported claims between the text and the figures, both in this version and in reference to the original version. We have also included further analysis into figure 2 to clarify how KAP-GFP fluorescence is altered in BCI treated cells. In addition, we have expanded the main text and discussion to include points previously made in the rebuttal and created a more focused section around how ciliogenesis is ultimately impacted by all of the processes described in the manuscript.

Reviewer #1 (Comments to the Authors (Required)):

The authors made significant changes and additions in this revision compared to the previous submission for which I served as reviewer #1 for Review Commons. As the authors summarized, this revision strengthened their interpretation and conclusion in several aspects, particularly the clarification of statistical significance and the assays involving double mutants. Additionally, the removal of several confusing or hard to interpret data sets allow the main observations to be the focus of this manuscript. I agree with the authors that this revision presents a much clearer and more concise story overall.

This revision continues to make seven major claims summarized in Figure 7. Based on the updated manuscript, below are my assessments on each main point of the paper.

1. BCI induced MAPK phosphorylation disrupts ciliogenesis.

While the single mutants do not show significant defect in ciliogenesis following deciliation by pH shock, two of the three double mutants partially phenocopy the BCI induced ciliogenesis phenotype. Consistently, the change in MAPK photophosphorylation was also most prominent in these two mutants. Given the background and scope of this manuscript, I would consider the interpretation supported with limitations.

2. BCI disrupts KAP-GFP protein synthesis during ciliogenesis and ciliary dynamics. The reported changes in Fig. 2 are consistent with a disruption in KAP-GFP dynamics. For reduced KAP-GFP expression, statistical significance directly comparing the third and fourth column in Fig 3C (similar to in 3D) is needed. A secondary, preferably a more sensitive method should be employed: for example, while the basal body fluorescence did not change significantly, did the overall fluorescence or the part in the cell body except basal bodies change? Or is there a major change in transcription/mRNA level? These might take several weeks to test depending on the technical limitations.

- We have included a statistical comparison of the 120 min timepoints in Fig 3C.
- We believe the regeneration western blot shown in Figure 3B is very clear evidence that KAP-GFP expression is indeed reduced in conjunction with the fluorescent data. We have this other data measuring cell body fluorescence,

but the data are highly variable and do not clarify our conclusions any further based on the basal body intensity.

- While we had initially thought about checking transcription, we did not pursue this experiment because regardless of what's happening in transcription, we see that protein expression is decreased as also evidenced by total protein expression shown in Fig 3B. While it would be interesting to know if transcription vs. translation is repressed as a result of this drug treatment, it does not change the fact that KAP-GFP that is present in the cell is still recruited to the basal bodies even though this protein expression is inhibited.

3. BCI disrupts NPHP4 expression and turnover at the transition zone.

The decrease in NPHP4 expression reported here is categorically different from the *nphp4* mutant (null) reported in Awata et al. 2014 and is marginally significant based on the authors' criteria. Additionally, the authors pointed out BCI "does not grossly disrupt the transition zone as a mechanism for altering ciliogenesis" (line 309). Therefore, I do not think the evidence provided provides satisfactory support to this point. The author may demonstrate a dosage-dependent impact of NPHP4 at the level demonstrated in this dataset or directly test that NPHP4 turnover is indeed altered due to BCI. Given these may take considerable effort and time, I would like to reiterate my previous comment (Major comment #4) about leaving this part out.

- We have mitigated our interpretation and language regarding this figure. We have changed the title of the section to: **BCI has minor effects on a transition zone protein but not on gross transition zone structure.**
- While Awata et al., 2014 shows data in light of a completely null mutant, it's interesting to see how big of an impact the lack of a presence of this protein has on the protein composition within the flagella. In addition, knockdown of this protein barely has an impact on ciliogenesis while it affects its regulatory processes. We cannot rule out that attenuated expression of NPHP4, though marginally but still significant, is not impacting protein entry into cilia.

4. BCI disrupts membrane trafficking.

The increase in actin puncta in the *mkp2 x mkp3* double mutant is particularly shocking given that all three single mutants showed reduction in actin puncta in previous submission (previous Sup Fig 3C). Similarly, the *mkp3 x mkp5* double mutant rescued both single mutants (again, from previous Sup Fig 3C). Can the authors provide any explanation or interpretation of such a difference?

The updated introduction about actin punta and ARPC4 are very helpful in orienting the interpretation. As long as the authors can provide satisfactory explanation about the discrepancy (?) above, the interpretation of this data is valid.

- One explanation is that, on the basis of our other study (Bigge et al., 2020), we propose a model where plasma membrane incorporation is required for cilium growth. Based on this model, in whatever way these mutants are preventing cilium growth, it is possibly causing a backup in actin puncta by having the membrane backed up from this pathway because there is nowhere for this plasma membrane to go. This may not be causing the puncta, but by virtue of

this process, may be leading to a backup of these puncta. *Mkp2 x mkp3* has no ciliary growth relative to *mkp2* or *mkp3* and we see that there is an increased amount of puncta where this pathway is backed up. Additionally, *mkp3 x mkp5* has no ciliary growth yet has increased puncta relative to *mkp3* or *mkp5* single mutants.

5. BCI disrupts cytoplasmic microtubule reorganization.

6. PTX stabilized microtubules do not rescue BCI induced ciliogenesis defects.

These two parts were and are the highlight of this manuscript. The changes in this revision materially improved this section and made the interpretation much easier to comprehend. One point that needs clarification is what the authors mean by the double mutants "could not fully reorganize their microtubules", as the changes from pre to 60m in SF 4B are not different between WT and double mutants.

- This analysis is correct. We have added the following statements to the discussion: "While the percentage of cells with intact microtubule arrays were reduced in double mutants ("pre" in WT cells vs "pre" in double mutants), microtubules in double mutants were able to reorganize to their pre-deflagellation levels in double mutants. While microtubule integrity may globally be affected by impaired MKP function, cytoplasmic microtubule reorganization following deflagellation is not affected to the degree seen for BCI treatment. This suggests the BCI has additional effects beyond what is recapitulated by these specific MKPs."
- We have also updated the results text to be more clear regarding the data: "We checked microtubule reorganization in regenerating DUSP6 ortholog double mutants and found that *mkp2 x mkp3* and *mkp3 x mkp5* mutants had fewer cells overall which maintained full cytoplasmic microtubules arrays, though these cells were able to reorganize microtubules to previous amounts within 60 minutes. This suggests a role for these ortholog mutants in regulating early cytoplasmic microtubule organization in the cell..."

Minor comments:

Line 238, unsupported claim: unclear if the decrease in fluorescence is either ciliary length dependent or significant

- Using the data in Figure 2C, we completed further comparisons and included these in Figure 2. Linear regression of the data shows that the lines are not significantly non-zero or that their slopes or intercepts are really different. However, total ciliary fluorescence in BCI treated cells was decreased as expected with shorter cilia. But, normalizing the fluorescence against the length of the cilium to generate the amount of fluorescence/ μm of cilia shows a nonsignificant difference in fluorescence. This suggests that the decrease in fluorescence is ciliary length dependent (fluorescence is decreasing proportionally with ciliary length), though the normalized fluorescence/micrometer of cilia is not significant. This also further supports that the shorter cilia, which do not have the increase in KAP-GFP fluorescence per micrometer of cilia, have a KAP-GFP entry defect.

- The text has been changed to “After a 2 hour treatment with BCI, cells with cilia maintained KAP-GFP fluorescence at their basal bodies regardless of BCI concentration (Fig 2B). However, at 30 μ M BCI, total KAP-GFP fluorescence in the cilia decreased (Fig 2C and D). Interestingly, the normalized fluorescence per micrometer of cilia was not significantly different between DMSO and BCI treated cells (Fig 2E) suggesting that KAP-GFP fluorescence proportionally decreases with ciliary length.”

Line 299, mis interpretation: no decrease of KAP-GFP at the basal bodies, see line line 277 and Fig 3D.

- The text has been changed to: “Unchanged KAP-GFP localization at the basal bodies but decreased frequency of KAP-GFP ciliary entry suggests that there could be a defect in the transition zone structure.”

Line 310, unsupported claim: protein turnover untested.

- We have removed “or protein turnover”

Line 331, unsupported claim (yet): at this point, the additive effect of BCI and BFA only exclude Golgi from potential membrane trafficking related targets of BCI.

- We have changed our concluding sentence to “This suggests that BCI excludes Golgi from potential membrane trafficking related targets of BCI (Fig 5A).”

Line453, misinterpretation: the data showed MTs organization defect induced by BCI does not contribute to the early ciliogenesis phenotype. The conclusion can't be broadened since it's only tested by rescuing with PTX.

- We have changed the text to “Our data also provides evidence that PTX stabilized cytoplasmic microtubule organization does not contribute to early ciliogenesis...”

Larisee Dougherty et al. explored how MAP kinases can regulate ciliary length using an unusual approach, focusing on a phosphatase inhibitor, BCI, that activates the MAP kinase. This study represents a new way of researching this topic and the provided data is of great value to the community. Given satisfactory response to the comments above, I consider this manuscript a strong fit for Life Science Alliance.

Reviewer #2 (Comments to the Authors (Required)):

The revised manuscript is greatly improved, and the authors successfully addressed many of the reviewer comments. The authors provided a complete and convincing rebuttal, however, some of this logic and language is not present in the final manuscript. To effectively convince the readers of the results, the authors should incorporate more points from their rebuttal in the main text. The authors did this in several instances. Two examples include expansion of the actin puncta and KAP-GFP dynamics

experiments/analysis. Here we outline the other instances where additional text or changes to the summary figure are necessary to clarify the results and interpretations.

MAJOR ISSUES

1. We appreciate that the manuscript includes references examining the off-target effects of BCI in zebrafish. This does not entirely address the possibility of *Chlamydomonas* specific off target effects of BCI. Please include language regarding how off target effects of BCI have not been directly explored in *Chlamydomonas* and to comment on why BCI was chosen over other pharmacological tools.

- We have changed text in the first results paragraph to: “Given the lengthening effects of ERK inhibition on ciliary length through U0126 (Avasthi et al., 2012), we wanted to test specificity of the pathway in ciliary regulation through activation of the pathway. BCI has previously been found to activate ERK1/2 upon inhibiting the dual specificity phosphatase/MAP kinase phosphatase, DUSP6/MKP3 in addition to DUSP1. BCI was selected as a tool due to the dramatic phenotype of completely blocked ciliary growth.”
- In the discussion, we have added “While BCI does not inhibit various other FGF inhibitors or related phosphatases including Cdc25B, PTP1B, and Dusp3/VHR in zebrafish, in *Chlamydomonas*, our various experiments identify effects of BCI not entirely recapitulated by single or even double MKP mutants. This suggests there are likely additional MKP or off-target effects of the drug.

2. It is possible that BCI could perturb the cell cycle, and therefore indirectly affect cilia assembly and disassembly. The authors begin to address this in the discussion, “BCI could arrest cells at the interphase checkpoint or induce cell cycle arrest which could... shift to assembly or disassembly”. Please expand on this limitation in the discussion.

- We have added the text in red:
- “Together, we have found mechanisms contributing to ciliary defects which could be the foundation of various ciliary diseases; the cilium acts as a major signaling hub in the cell to regulate cellular activities. When this structure cannot form and function properly on ciliated cells, important signals cannot be transmitted throughout the cell. Cells normally resorb their cilia for division and then reassemble them during G1/G0. BCI could arrest cells at the interphase checkpoint or induce cell cycle arrest which could inhibit protein synthesis to shift ciliary assembly to disassembly. As shown in Figure 1F, the double ortholog mutants *mkp2 x mkp3* and *mkp3 x mkp5* do not respond to BCI with an increase in pMAPK expression unlike *mkp2 x mkp5* and the ortholog single mutants. It has previously been shown that while pMAPK is not necessary for cell division in *Drosophila*, it is required for cell growth (Majumdar et al., 2010). Without this signal being generated, these cells are either stuck in cell division which does not allow for ciliogenesis, or in the case of BCI treatment, cells are continually signaled to grow and this may ultimately lead to ciliary resorption. Constant ERK activation can also lead to negative feedback to regulate this signal for cell cycle progression (Fritsche-Guenther et al., 2011). The ERK pathway has not been fully elucidated in *Chlamydomonas*, but it is possible that these similar mechanisms are in place for MAPKs. Further work is needed to identify the

precise cause of completely inhibited cilium growth from zero length. Regardless, the insight provided here can help to provide a basis for understanding more broad mechanisms for how ciliopathies or manipulated cilia can drastically alter the fate of the cell.”

3. In response to our question "why is the cilia length and regeneration phenotype different in *Chlamydomonas* compared to RPE1 cells" the authors provide a clear response in the rebuttal. Please include this language in the manuscript with the hTERT-RPE1 results.

- In this updated version, we no longer have the data present that describes an alternative regeneration phenotype, so this description no longer applies to the data we are directly showing. Presented in the manuscript is a similar phenotype to RPE1 cells where there is a dose-dependent shortening effect in addition to a spike in pMAPK expression following BCI treatment similar to what occurs in *Chlamydomonas*.

4. To address our critique that protein synthesis is not directly examined, the authors state in their rebuttal that they "use KAP-GFP readout as a proxy for protein synthesis". This language must be included in the manuscript to clarify the interpretation of the results in Fig. 3. While there is some evidence that protein degradation is not impacted (an n=1 is incomplete), we ask the authors to moderate the conclusion that protein synthesis is decreased throughout the manuscript. This result does not eliminate alternative possibilities, and this should be addressed in the discussion.

- Given the requests for more specific language regarding KAP-GFP expression in the first revision and the complete removal of the cycloheximide experiment where a majority of our claims regarding general protein synthesis originated, the text has already been adjusted to reflect this comment. If we are missing places in the text where this is located, we request that these be directly listed.

5. Disruptions to cilia assembly and disassembly are not mutually exclusive. The text needs expanding to include the possibility that BCI could impact disassembly, as disassembly was not tested.

- We have added “While effects on disassembly remains a possibility, the striking inability to increase from zero length upon deciliation and the effects on anterograde IFT through the TIRFM assays suggest an affect on assembly.” To the end of the first paragraph of the discussion.

6. We appreciate that the authors changed the language around Kinesin-2 to reflect the direct observations of KAP-GFP. The authors also state in the rebuttal that "Because KAP-GFP is a cargo adaptor protein and interacts with IFT complexes and other cargoes, defects in this protein can have a wide range of implications". Please include this text in the manuscript to expand on the interpretation of Fig. 3.

- We have added “The KAP subunit is a cargo adaptor protein which is known to interact with IFT complexes and other cargoes for anterograde transport. Defects in this adaptor can have a wide range of implications for ciliogenesis.”

to the first paragraph under the header “BCI disrupts ciliary KAP-GFP dynamics”

7. The authors do a great job addressing individual points in the rebuttal, but this not reflected in the discussion section. The discussion still falls short on synthesizing the new results. As written, the discussion summarizes all the results as independent processes. Instead, the discussion should address how these disrupted processes all connect to ciliogenesis, which is the predominant phenotype following BCI treatment. Similarly, the final figure should reflect how the results all connect to ciliogenesis instead of disparate consequences of BCI treatment. Currently, the boxes in the figure overstate the results (ex: transition zone protein composition is an overstatement of NPHP4 protein expression) and expanded language in the figure legend or directly in the figure is necessary to reduce these broad conclusions.

- We have added a statement to the discussion: “By dissecting various processes related to stages of ciliogenesis (including ciliary protein synthesis, intracellular membrane and protein trafficking to basal bodies, ciliary gating, and IFT dynamics), we find it likely that activation of the ERK pathway via MKP inhibition has multiple touchpoints that ultimately affect ciliogenesis. The broad range of these effects is somewhat expected considering the quantity and diversity of known downstream ERK targets within cells.”
- In addition, we have added a couple of clarifying statements to generate a response more centered around how these processes connect to ciliogenesis:
 - Regarding KAP-GFP protein synthesis: “Our data also provides evidence that PTX-stabilized cytoplasmic microtubule organization does not contribute to early ciliogenesis, though BCI inhibits its reorganization (**Fig 6**). Prevented microtubule polymerization in the presence of BCI following deciliation likely traps KAP-GFP at the basal bodies post deciliation (**Fig 3A**) as well as other proteins required for ciliogenesis at their other locations which may ultimately lead to ciliary shortening or inhibited regeneration. Lack of these protein highways could also prevent mRNA from localizing to its intended area for adequate protein synthesis of precursor proteins required for ciliogenesis. Constitutive ERK activation could also induce negative feedback loops that prevent transcription as well which would inhibit mRNA formation.”
 - Regarding membrane trafficking: “Previous work has shown that inhibition of Golgi-derived membrane induces ciliary shortening through Golgi collapse. The epistatic ciliary shortening in BFA and BCI together suggests that BCI may create an additive effect on the Golgi that may disrupt protein and membrane trafficking for ciliogenesis from a source separate from the Golgi which is supported by the defects in endocytosis as measured by actin puncta.”
 - And we have included text described in Major Issue #2 regarding MAPK activation: “Without this signal being able to be generated, these cells are either stuck in cell division which does not allow for ciliogenesis, or in the case of BCI treatment, cells are signaled to grow and this ultimately leads to ciliary resorption. In the presence of this drug, cells are stuck preparing

for growth and cell division and likely cannot regenerate cilia. Constant ERK activation can lead to negative feedback to regulate this signal for cell cycle progression”

- As mentioned in the initial rebuttal, we have formed a summary model based on what holds true to the data. We do not have data to extrapolate beyond these means; while we know these processes are happening following pMAPK activation and ciliary shortening, we do not know at what point these processes are occurring relative to each other or what factors are occurring in between to provide a more chronological summary than what is currently depicted. We would love to extrapolate beyond this model for how these pieces are likely fitting together, but this simply wouldn't be true. We will not generate a model that is not supported based on the data. We have updated the current depiction to be more specific to the data by replacing the headings:
 - “kinesin-2 ciliary dynamics” to “KAP-GFP ciliary dynamics”
 - “Protein synthesis” to “KAP-GFP protein synthesis”
 - “Transition zone protein composition” to “NPHP4 Localization at the Transition Zone”

MINOR ISSUES

1. While the experiment with cycloheximide was removed, the cycloheximide text is still present in the discussion, methods, and references.

- We have removed the cycloheximide text from the discussion, methods, and references.
- From the discussion, we removed:
 - “The fact that cilia can grow in cycloheximide to half length suggests that there is enough KAP-GFP and total ciliary protein present at the ciliary base for ciliary incorporation, but BCI immediately prevents incorporation of the precursor pool.”
 - “There is normally constant turnover of tubulin and ciliary proteins which is normally replaced and because BCI inhibits protein synthesis (**Fig 3**), cilia do not have the supplies to keep growing cilia and thus we see ciliary shortening (Song & Dentler, 2001; Stephens, 1997). While the protein synthesis defect does not explain why there is complete and immediate inhibition of ciliogenesis in regenerating cells, it can help explain why cilia cannot grow over time with MAPK activation.”
- From the methods: “For double regenerations, cells were allowed to grow for 1 hour in 10 µg/mL cycloheximide (Sigma, C1988-1G) and then pH shocked a second time, resuspended in new TAP, and grown with or without drugs.”
- From references:
 - Song & Dentler, 2001
 - Stephens, 1997

2. Line 299 states "Decreased KAP-GFP expression at the basal bodies..." which contradicts the result in Fig. 3D that there is no change in KAP-GFP expression at the basal bodies.

- We thank both reviewers for catching this. We have updated the text accordingly.
3. Line 296 should write "Given the effect of BCI on KAP-GFP ciliary entry,..."
 - We have updated to the text to "Given the effect of BCI on KAP-GFP ciliary entry..."
 4. Remove "and hTERT-RPE1" from the Figure 1 legend since those results are now in supplemental.
 - We have removed "and hTERT-RPE1" from the figure 1 legend title.

March 1, 2023

RE: Life Science Alliance Manuscript #LSA-2023-01899R

Dr. Prachee Avasthi
Geisel School of Medicine at Dartmouth College
74 College St
HB 7200
Hanover, NH 03755

Dear Dr. Avasthi,

Thank you for submitting your revised manuscript entitled "ERK activation regulates ciliogenesis through disrupting ciliary motors, gating and microtubules". We would be happy to publish your paper in Life Science Alliance pending final revisions necessary to meet our formatting guidelines.

Along with points mentioned below, please tend to the following:
-please add the author contributions to the main manuscript text

A. FINAL FILES:

B. MANUSCRIPT ORGANIZATION AND FORMATTING:

Sincerely,

March 3, 2023

RE: Life Science Alliance Manuscript #LSA-2023-01899RR

Dr. Prachee Avasthi
Geisel School of Medicine at Dartmouth College
74 College St
HB 7200
Hanover, NH 03755

Dear Dr. Avasthi,

Thank you for submitting your Research Article entitled "ERK activation regulates ciliogenesis through disrupting ciliary motors, gating and microtubules". It is a pleasure to let you know that your manuscript is now accepted for publication in Life Science Alliance. Congratulations on this interesting work.

DISTRIBUTION OF MATERIALS:

Again, congratulations on a very nice paper. I hope you found the review process to be constructive and are pleased with how the manuscript was handled editorially. We look forward to future exciting submissions from your lab.

Sincerely,
